# Universality of the SAT-UNSAT (jamming) threshold
# in non-convex continuous constraint satisfaction problems

**Silvio Franz[1*], Giorgio Parisi[2,3], Maksim Sevelev[1],**
**Pierfrancesco Urbani[4] and Francesco Zamponi[5]**

**1** LPTMS, CNRS, Univ. Paris-Sud, Université Paris-Saclay, 91405 Orsay, France
**2** Dipartimento di Fisica, Sapienza Università di Roma, P.le A. Moro 2, I-00185 Roma, Italy
**3** INFN, Sezione di Roma I, Nanotec – CNR, P.le A. Moro 2, I-00185 Roma, Italy
**4** Institut de physique théorique, Université Paris Saclay, CNRS, CEA,
F-91191 Gif-sur-Yvette, France
**5** Laboratoire de physique théorique, Département de physique de l'ENS,
École normale supérieure, PSL Research University, Sorbonne Universités,
UPMC Univ. Paris 06, CNRS, 75005 Paris, France

* silvio.franz@lptms.u-psud.fr

## Abstract

**Random constraint satisfaction problems (CSP) have been studied extensively using statistical physics techniques. They provide a benchmark to study average case scenarios instead of the worst case one. The interplay between statistical physics of disordered systems and computer science has brought new light into the realm of computational complexity theory, by introducing the notion of *clustering* of solutions, related to replica symmetry breaking. However, the class of problems in which clustering has been studied often involve discrete degrees of freedom: standard random CSPs are random $K$-SAT (*aka* disordered Ising models) or random coloring problems (*aka* disordered Potts models). In this work we consider instead problems that involve continuous degrees of freedom. The simplest prototype of these problems is the perceptron. Here we discuss in detail the full phase diagram of the model. In the regions of parameter space where the problem is non-convex, leading to multiple disconnected clusters of solutions, the solution is critical at the SAT/UNSAT threshold and lies in the same universality class of the jamming transition of soft spheres. We show how the critical behavior at the satisfiability threshold emerges, and we compute the critical exponents associated to the approach to the transition from both the SAT and UNSAT phase. We conjecture that there is a large universality class of non-convex continuous CSPs whose SAT-UNSAT threshold is described by the same scaling solution.**

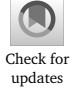

# 1   Introduction

The exact description of the glassy phases of high-dimensional sphere systems and the discovery that universal predictions at jamming match finite dimensional observations [1] has renewed the interest in the statistical physics of random constraint satisfaction problems (CSP) with continuous variables. In CSPs, one seeks assignments of a set of $N$ variables that satisfy a system of constraints. Sphere systems clearly belong to this class: one needs to find the positions of $N$ spheres inside a box with the conditions that the spheres do not overlap. Particularly interesting in a statistical physics perspective is the case where the constraints are taken at random from some ensemble [2–4]. In that case, quite generically in the limit of large systems one observes a phase transition as the density of constraints is increased passing from a satisfiable (SAT) phase where admissible configurations exist to an unsatisfiable (UNSAT) phase, where the minimum number of unsatisfied constraints is larger than zero. This SAT/UNSAT threshold is clearly analogue to the jamming transition in soft-spheres [5,6], that separates the low density region where spheres do not overlap from the high density one where no-overlap configurations do not exist. The jamming transition in spheres has highly universal features, with exponents that appear to be independent of the dimension and of the protocol used to produce the jammed configurations [1,5–12]. While some exponents, like e.g. the ones relating the number of contacts or the pressure to the packing fraction in the UNSAT phase, take simple semi-integer values [11], other exponents, e.g. the ones describing the distributions of forces and the interparticle distances have non trivial, presumably non-rational values [13].

In order to understand the origin of this universality, it is important to study the SAT/UNSAT transition in different CSP models. A crucial ingredient for jamming is the continuous nature of variables. Jamming is the point where the volume of the set of solutions to the problem continuously shrinks to zero, and in its vicinity scaling laws can emerge. To this aim, in [14] it was suggested to study the random perceptron problem [15] as a prototype of a CSP with continuous variables, generalized to a region of non-convex optimization. It was found that the nontrivial criticality and universality of the SAT/UNSAT transition point was associated to non-convexity. In the convex regime jamming is reached from a liquid, ergodic phase and it is hypostatic and non-critical. In the non-convex regime jamming is reached from a glassy phase, it is critical and in the same universality class of spheres. This led to the conjecture that it exists a large set of continuous CSP that belong to the same universality class. The perceptron emerges therefore as the simplest continuous CSP where glassy phenomena and jamming can be studied. This paradigm has been fruitfully applied to study the vibrational spectrum of glasses at low temperatures. In [16] the spectrum of the Hessian matrix of the energy minima in the UNSAT phase of the perceptron has been computed showing that it captures essential features of the vibrations of low temperature glasses. Furthermore, in [17] it has been shown how to study systematically the free energy landscape of the model using a Thouless-Anderson-Palmer approach [18] to obtain, in particular, the vibrational spectrum in the SAT phase. In [19], the avalanches characterizing the glassy phase around jamming have also been studied. Thanks to these studies, the non-convex perceptron now emerges as the simplest model that captures, at the mean field level, all the most important features of the glass and jamming transitions.

The scope of this paper is to give a detailed account of the space of solutions of the random perceptron model. In particular we carefully study the scaling behavior close to jamming, coming both from the SAT and UNSAT phase. The paper is organized as follows. In Sec. 2 we give a general formulation of continuous CSPs, we discuss the properties of the SAT-UNSAT transition, and we briefly discuss the case of sphere packings to motivate some denominations that are used throughout the paper. In Sec. 3 we define the random perceptron model, we introduce the replica method to solve it, and we give the main equations needed for its study.

In Sec. 4 we discuss the zero-temperature phase diagram of the model, and in Sec. 5 we completely characterize the SAT/UNSAT transition (jamming) line and its critical properties. Finally, we present concluding remarks and perspectives for future work.

## 2 Continuous constraint satisfaction problems

### 2.1 Thermodynamic free energy, and the space of solutions

In an abstract form continuous CSPs can be formulated in the following way: find a $N$-dimensional vector $\vec{X} = \{X_i\}_{i=1\cdots N} \in \mathbb{R}^N$ that satisfies the set of $M$ constraints

$$h_\mu(\vec{X}) > 0 , \qquad \mu = 1, ..., M . \tag{1}$$

The constraints are specified by some real functions $h_\mu(\vec{X}) : \mathbb{R}^N \to \mathbb{R}$, which can be either deterministic or random (i.e. they may contain some quenched disorder). One can associate this problem to an optimization one, defining a Hamiltonian function taking positive values if at least one constraint is not satified and zero if all the constraints are satisfied. There is a large choice for such a Hamiltonian; here in analogy with harmonic soft spheres [5] we choose

$$H[\vec{X}] = \sum_{\mu=1}^{M} v(h_\mu(\vec{X})) = \frac{1}{2} \sum_{\mu=1}^{M} h_\mu(\vec{X})^2 \theta(-h_\mu(\vec{X})) , \qquad v(h) = \frac{1}{2} h^2 \theta(-h) . \tag{2}$$

Other choices of $v(h)$ can be considered, provided $v(h > 0) = 0$ and $v(h < 0) > 0$. The analysis of the space of solutions can be performed, following Derrida and Gardner [15], from the study of the partition function

$$Z = \int \mathscr{D}\vec{X} e^{-\beta H[\vec{X}]} = \int \mathscr{D}\vec{X} e^{-\frac{\beta}{2} \sum_{\mu=1}^{M} h_\mu(\vec{X})^2 \theta(-h_\mu(\vec{X}))} , \tag{3}$$

where the measure $\mathscr{D}\vec{X}$ may include some additional normalization constraint.

One is typically interested in the limit $N \to \infty$, with the number of constraints $M$ scaled appropriately to have a non-trivial limit. In this limit, a sharp SAT-UNSAT phase transition emerges [2–4], and can be characterized by looking at the zero temperature limit of the partition function. In the SAT phase, the ground state energy is equal to zero, and the partition function reduces to the volume of the space of satisfying assignments (whose logarithm is the entropy of solutions). The corresponding homogeneous measure on the space of the solutions gives identical weights to all configurations that satisfy the constraints. In the UNSAT phase, the ground state energy is non-zero and the partition function is dominated by the ground state configurations.

One is usually interested in the free energy per particle,

$$\mathrm{f} = -\frac{T}{N} \overline{\log Z} , \tag{4}$$

where the overline indicates an average over the quenched disorder, if it is present in the constraints. This quantity has a finite limit for $N \to \infty$ and allows one to extract easily all the thermodynamic information; moreover, in presence of quenched disorder, the free energy per particle is usually self-averaging [20], i.e. its fluctuations due to the disorder vanish for $N \to \infty$. In general, $\mathrm{f} = e - Ts$, where $e$ is the thermodynamic energy and $s$ the thermodynamic entropy. In particular, in the SAT phase, when $T \to 0$, $e \to 0$ and $s$ has a finite limit; as a result $-\beta \mathrm{f} \to s$. On the contrary, in the UNSAT phase, when $T \to 0$, $e$ has a finite limit, and $Ts \to 0$, and as a result $\mathrm{f} = e$.

Note that besides the thermodynamic (or "equilibrium") SAT-UNSAT transition, defined as above from the partition function in Eq. (3), one can study many other similar transitions that happen out of equilibrium. Indeed, in the SAT phase at high enough density of constraints, the equilibrium state of the system is a collection of distinct thermodynamic states [4]. One can restrict the study to one of these states (also called the "state following" formalism [21–24]) and study the SAT-UNSAT transition of the Boltzmann-Gibbs measure restricted to that particular state. It has been shown for sphere systems that this does not change the critical properties of the transition [24], hence in the following we restrict our study to the equilibrium setting.

## 2.2 Distribution of gaps

Besides the free energy and its derived quantities, another interesting observable, in particular in the context of jamming, is the probability distribution of "gaps". In fact, it has been shown that this quantity encodes important information about the marginal stability of jammed configurations that we are going to discuss below [7].

A gap variable is just a constraint function $h_\mu(\vec{X})$ – the name "gap" originates from the fact that $h_\mu = 0$ corresponds to a constraint being on the verge of becoming unsatisfied (or a "contact"), and the value of $h_\mu$ is thus the distance ("gap") to this configuration. The gap probability distribution $\rho(h)$ is defined as

$$\rho(h) = \overline{\langle \hat{\rho}(h) \rangle}, \qquad \hat{\rho}(h) = \frac{1}{M} \sum_{\mu=1}^{M} \delta(h - h_\mu(\vec{X})), \tag{5}$$

where the brackets denote a thermodynamical average, while the overline denotes an average over quenched disorder, if present. From $\rho(h)$ one can derive

$$z \equiv \int_{-\infty}^{0} \mathrm{d}h \, \rho(h), \tag{6}$$

which is the average fraction of unsatisfied constraints, or fraction of contacts. To compute the distribution of gaps, we note that the partition function can be written as

$$Z = \int \mathscr{D}\vec{X} e^{-\beta H[\vec{X}]} = \int \mathscr{D}\vec{X} e^{M \int \mathrm{d}h \hat{\rho}(h)[-\beta v(h)]}. \tag{7}$$

Hence, because $-\beta f = \overline{\log Z}/N$, we get

$$\frac{\mathrm{d}f}{\mathrm{d}v(h)} = \frac{1}{N} \frac{\mathrm{d}\overline{\log Z}}{\mathrm{d}[-\beta v(h)]} = \alpha \overline{\langle \hat{\rho}(h) \rangle} = \alpha \rho(h). \tag{8}$$

In the SAT phase, there are no unsatisfied constraints, and $\rho(h) = 0$ for $h < 0$, so that $z = 0$, while in the UNSAT phase $\rho(h)$ is non-zero for $h < 0$ and $z > 0$. A particularly important quantity is the limit value of $z$ when one approaches the SAT-UNSAT transition coming from the UNSAT phase where $z > 0$. This limit is usually strictly positive, hence $z$ jumps discontinuously at the transition. If this limiting value is such that the number of unsatisfied constraints is exactly equal to the number of degrees of freedom, i.e. $Mz = N$, then the system is said to be *isostatic* [6]. We can therefore define an *isostaticity index* $c = (M/N)z$, which is equal to 1 if the system is isostatic. More generally, the system is said to be hypostatic, isostatic or hyperstatic whenever the total number of violated constraints is less ($c < 1$), equal ($c = 1$) or higher ($c > 1$) than the total number of degrees of freedom.

We will be particularly interested in observables of the form

$$\mathcal{O}[\vec{X}] = M^{-1} \sum_{\mu=1}^{M} \mathcal{O}(h_\mu)\theta(-h_\mu), \tag{9}$$

that are functions of the negative gaps. We define the thermodynamic and disorder average

$$[\mathcal{O}(h)] = \overline{\langle \mathcal{O} \rangle} \equiv \frac{1}{M} \sum_{\mu=1}^{M} \overline{\langle \mathcal{O}(h_\mu)\theta(-h_\mu) \rangle} = \int dh \rho(h) \mathcal{O}(h)\theta(-h) . \tag{10}$$

Special cases of this class of observables are the thermodynamic energy $e = \overline{\langle H[\vec{X}] \rangle}/N$, the "pressure" $p$, and the fraction of contacts:

$$z = [1] , \qquad p = -[h] , \qquad e = \alpha[\nu(h)] = \alpha[h^2]/2 . \tag{11}$$

### 2.3 Some consequences of isostaticity: force distribution and soft modes

We now formulate in our abstract continuous CSP setting some well-known consequences of isostaticity [8,10], i.e. the condition that the number of unsatisfied constraints is equal to the number of degrees of freedom, and $c = 1$. We omit for notational simplicity the $\vec{X}$-dependence of the gaps $h_\mu$ and we define, with respect to the Hamiltonian given in Eq. (2):

$$\begin{aligned}
F_i &= -\frac{dH}{dX_i} = \sum_{\mu=1}^{M} [-h_\mu\theta(-h_\mu)]\frac{dh_\mu}{dX_i} = \sum_{\mu\in\mathscr{C}} \mathscr{S}_{\mu i} f_\mu , \\
f_\mu &= -h_\mu\theta(-h_\mu) , \\
\mathscr{S}_{\mu i} &= \frac{dh_\mu}{dX_i} ,
\end{aligned} \tag{12}$$

where the set of "contacts" $\mathscr{C} = \{\mu : h_\mu < 0\}$ is the set of unsatisfied constraints, of size $|\mathscr{C}| = Mz = Nc$, the matrix $\mathscr{S}$ has dimension $Mz \times N$, and the "contact force" vector $\vec{f}$ has dimension $Mz$ because we consider only the non-zero components $f_\mu$. We therefore have $\vec{F} = \mathscr{S}^T \vec{f}$. At zero temperature, we are especially interested in minima of the Hamiltonian, which correspond to vanishing "total forces" $\vec{F} = \mathscr{S}^T \vec{f} = \vec{0}$. Furthermore, one is often interested in the small harmonic vibrations around a minimum, which are described by the Hessian matrix

$$\mathscr{H}_{ij} = \frac{d^2 H}{dX_i dX_j} = \sum_{\mu\in\mathscr{C}} \frac{dh_\mu}{dX_i}\frac{dh_\mu}{dX_j} + \sum_{\mu\in\mathscr{C}} h_\mu \frac{d^2 h_\mu}{dX_i dX_j} . \tag{13}$$

It is particularly interesting to consider the above structure in the jamming limit, i.e. at the SAT-UNSAT transition. Note that "contacts", i.e. constraints for which $h_\mu < 0$ in the UNSAT phase close to the transition, must then become marginally satisfied, i.e. $h_\mu = 0$, right at the transition. Moreover, the average contact force is proportional to the pressure, $|\mathscr{C}|^{-1}\sum_{\mu\in\mathscr{C}} f_\mu = -[h]/[1] = p/[1]$, which indeed vanishes at jamming. For contacts, one can thus define scaled forces $f_\mu^s = [1]f_\mu/p$, that remain finite at the jamming transition. These scaled forces still satisfy the condition $\mathscr{S}^T \vec{f}^s = 0$, where the matrix $\mathscr{S}$ is calculated at the jamming point. Although by definition both $\mathscr{S}$ and $\vec{f}$ are fully determined by the configuration $\vec{X}$, one can ask in general how many solutions $\vec{f}$ to $\mathscr{S}^T \vec{f} = \vec{0}$ can be found, for fixed $\mathscr{S}$. Because this is a linear equation, for hypostatic systems ($c < 1$) there are in general no solutions, while for hyperstatic systems ($c > 1$) there are in general $N(c-1)$ solutions, and for isostatic systems ($c = 1$) there is a single solution. Therefore, for an isostatic system, the vector of scaled forces

must necessarily coincide with the unique solution of $\mathscr{S}^T \vec{f}^s = \vec{0}$. This is particularly useful for numerical calculations, because the force vector vanishes at jamming and it is therefore difficult to evaluate the scaled forces with high precision, while the matrix $\mathscr{S}$ remains finite and determining its unique zero mode requires much lower precision [10]. Moreover, while in the SAT phase in principle the contact forces vanish, one can still define effective contact forces by following the procedure outlined in [25]. Then, one finds that the scaled contact forces also converge to the zero mode of $\mathscr{S}$ when approaching jamming from the SAT phase. Finally, one can consider the Hessian matrix at jamming. The second term in Eq. (13) vanishes because $h_\mu = 0$ for contacts, and therefore $\mathscr{H} = \mathscr{S}^T \mathscr{S}$. From this one can deduce that in general $\mathscr{H}$ has $N(1-c)$ zero modes for hypostatic systems, while it has no zero modes for hyperstatic systems; therefore at jamming, which separates the two situations, one finds a large number of very small eigenvalues of the Hessian ("soft modes") [8].

We have seen in this section that, for general continuous CSPs, isostaticity at jamming implies that the contact forces are fully determined by the matrix $\mathscr{S}$, that also determines the soft modes of the Hessian. This mathematical structure has many other interesting consequences. We will not provide further details on these aspects. The interested reader can consult Ref. [10, Supplementary Information] for a detailed review of the properties of $\mathscr{S}$ in the context of spheres, Ref. [16] for a study of the Hessian in the perceptron, and Ref. [7, 8] for a discussion of soft modes in sphere packings.

## 2.4 Sphere packing as a constraint satisfaction problem

In order to motivate the interest for the observables discussed above (and their names), we discuss here more explicitly the sphere packing problem and its formulation as a continuous CSP. The $d$-dimensional sphere packing problem is indeed a special case of the general setting discussed above. One considers $n$ points in a $d$ dimensions volume, $\vec{x}_i \in V \subset \mathbb{R}^d$, $i = 1, \cdots, n$, hence the total number of degrees of freedom is $N = dn$. There are $M = n(n-1)/2$ constraints corresponding to all possible distinct particle pairs $\mu = \langle i, j \rangle$ (e.g. with $i < j$), of the form

$$h_\mu(\vec{X}) = |\vec{x}_i - \vec{x}_j| - \sigma_{ij} , \qquad \sigma_{ij} = \frac{\sigma_i + \sigma_j}{2} . \tag{14}$$

In this case, $\sigma_i$ can be interpreted as the diameter of particle $i$, and then $h_{\langle i,j \rangle}$ is precisely the physical gap between particles $i, j$. In particular, if $h_{\langle i,j \rangle} < 0$, then the two particles $i, j$ overlap and thus feel a repulsive interaction. This justifies the name "gap" given to $h_\mu$, and the name "fraction of contacts" given to $z$. Note that the fraction of contacts defined in Eq. (6) is normalized to the total number of constraints; in particle systems, it is customary to consider instead the average number of contacts per particle,

$$z_p = \frac{2Mz}{n} = \frac{2Nc}{n} = 2d\,c . \tag{15}$$

Hence, an isostaticity index $c = 1$ corresponds to $z_p = 2d$ contacts per particle. The Hamiltonian $H[\vec{X}]$ in Eq. (2), with the choice $v(h) = h^2 \theta(-h)/2$, is precisely the Hamiltonian of a system of soft harmonic repulsive spheres [5, 6]; the partition function in Eq. (3) is the thermodynamical canonical partition function of the model, and f the associated free energy per particle. The distribution of gaps is simply related to the "radial distribution function" of the particle system, and the average gap is related to the pressure. Finally, $f_\mu$ defined in Eq. (12) is precisely the modulus of the contact force associated to the particle pair $\langle ij \rangle$, and the matrix $\mathscr{S}$ encodes the network of contacts in the particle packing [10]. In this case, one is then interested in the limit $n, |V| \to \infty$ with fixed density $n/|V|$.

Note that in this case, in the SAT phase at $T \to 0$, the Boltzmann-Gibbs measure becomes a uniform measure over all the configurations satisfying the non-overlapping constraint, which

coincides with the equilibrium Boltzmann-Gibbs distribution of a system of hard spheres. In the UNSAT phase (which cannot exist for hard spheres) there are overlaps and, at zero temperature, one has instead a mechanically stable assembly of soft repulsive spheres.

In the sphere packing problem, at least when $\sigma_{ij} = \sigma$ (all particles are identical), there is an additional complication because, even within the SAT phase, the system can form a *crystal*, a phase in which the particles are arranged over a regular lattice in $\mathbb{R}^d$. Crystals are usually denser than disordered arrangements, and therefore the SAT-UNSAT transition usually happens within the crystal phase, although the situation is somewhat uncertain in large dimensions [26]. In this case, the SAT-UNSAT transition (also called "close packing" point) has very different properties with respect to the same transition in disordered assemblies. Yet, one can restrict to the study of disordered configurations and in that case the SAT-UNSAT transition (also called "jamming transition" or "random close packing" in this case) has robustly universal critical properties [6, 27].

## 3 Definition of the model and basic equations

Following [14], in this paper we want to study the properties of the simplest continuous CSP that exhibits a phenomenology similar to that of spheres (once one restricts to their disordered phase). While spheres do not have quenched disorder (the constraints are deterministic functions), the perceptron model has quenched disorder. The presence of quenched disorder eliminates any possible crystal phase and one can then study the equilibrium properties of the model in the disordered phase. Before getting into the study of the phase diagram, and its critical properties at the SAT-UNSAT transition, in this section we give the basic definition of the model and the main equations needed for its study.

### 3.1 Definition of the model and free energy

The perceptron model, on which we concentrate in the rest of this paper, is defined by the linear functions

$$h_\mu(\vec{X}) = \frac{1}{\sqrt{N}} \vec{X} \cdot \vec{\xi}^\mu - \sigma, \tag{16}$$

with the normalization $\vec{X} \cdot \vec{X} = N$. The vectors $\vec{\xi}^\mu$, that following the neural network literature [15] we call "patterns", have components $\xi_i^\mu$ which are independent Gaussian variables of zero average and unit variance. The partition function reads

$$Z = \int \mathscr{D}\vec{X} e^{-\beta H[\vec{X}]} = \int \mathscr{D}\vec{X} \prod_{\mu=1}^M \left[ \int \mathrm{d}r_\mu e^{-\beta v(r_\mu - \sigma)} \delta\left(r_\mu - N^{-1/2}\vec{X} \cdot \vec{\xi}^\mu\right) \right], \tag{17}$$

where the measure $\mathscr{D}\vec{X}$ contains the constraint $\vec{X} \cdot \vec{X} = N$, i.e. it is the uniform measure on the $N$-dimensional sphere of radius $\sqrt{N}$. For positive $\sigma$ the model can be interpreted as a classifier of the random patterns $\vec{\xi}^\mu$, and defines a convex optimization problem. For negative $\sigma$ the model cannot be interpreted as a classifier. It still is a legitimate CSP, but it is non-convex. It has been observed in [14] that this is the interesting regime to describe jamming and glassy phases. In both cases one considers the thermodynamic limit $N, M \to \infty$ for values of $\alpha = M/N$ fixed.

The disorder average of the free energy can be computed via the replica method:

$$\mathrm{f} = -\frac{T}{N} \overline{\log Z} = -\frac{T}{N} \lim_{n \to 0} \partial_n \overline{Z^n} \,. \tag{18}$$

The free energy f is obtained through standard manipulations [20] from the expression of $Z^n$ for integer $n$. At the end of a computation sketched in Appendix A, the free energy can be written as a saddle point over the $n \times n$ "replica overlap matrix"

$$Q_{ab} = \frac{1}{N} \overline{\langle \vec{X}_a \cdot \vec{X}_b \rangle} \, , \tag{19}$$

where $\vec{X}_a$ are replicas of the configurations of the system. Notice that the spherical contraint on the $\vec{X}$ implies $Q_{aa} = 1$. The explicit expression for the free-energy is

$$\text{f} = -T \lim_{n \to 0} s.p. \partial_n S(Q) \, , \tag{20}$$

where $s.p.$ denotes the saddle point over the values $Q_{ab}$ and

$$S(Q) = \frac{1}{2} \log \det Q + \alpha \log \left( e^{\frac{1}{2} \sum_{ab}^n Q_{ab} \frac{\partial^2}{\partial h_a \partial h_b}} \prod_{a=1}^n e^{-\beta v(h_a)} \Big|_{h_a = -\sigma} \right) \tag{21}$$

is the *replicated free energy*. The saddle point equation for $Q_{ab}$, $a \neq b$, is thus

$$0 = \left[ Q^{-1} \right]_{ab} + \alpha \beta^2 \langle v'(h_a) v'(h_b) \rangle \tag{22}$$

where we have defined

$$\langle O(h_a, h_b, \ldots) \rangle = \frac{e^{\frac{1}{2} \sum_{ab}^n Q_{ab} \frac{\partial^2}{\partial h_a \partial h_b}} O(h_a, h_b, \ldots) \prod_{a=1}^n e^{-\beta v(h_a)} \Big|_{h_a = -\sigma}}{e^{\frac{1}{2} \sum_{ab}^n Q_{ab} \frac{\partial^2}{\partial h_a \partial h_b}} \prod_{a=1}^n e^{-\beta v(h_a)} \Big|_{h_a = -\sigma}} \, . \tag{23}$$

Finding the solution of such kind of equations is an extremely difficult task. Furthermore, once the solution is found a sensible analytic continuation down to $n \to 0$ must be taken, which is an additional complication. The solution to both difficulties is provided by the use of hierarchical matrices. Here we do not review in detail this construction, which can be found in several reviews, e.g. [20].

## 3.2 Hierarchical ansatz for the saddle point solution

The general solution of Eq. (23), in the limit $n \to 0$, is given in terms of a continuous Parisi hierarchical matrix. In this case the matrix $Q_{ab}$ is parametrized by a function $q(x)$ in the interval $x \in [0, 1]$ whose general form is plotted in Fig. 1. For a hierarchical matrix, the first term of $S(Q)$ reads [28]

$$\lim_{n \to 0} \frac{1}{n} \log \det Q = \log(1 - q_M) + \frac{q_m}{\lambda(0)} + \int_0^1 dx \, \frac{\dot{q}(x)}{\lambda(x)} \, , \tag{24}$$

where the dot denotes differentiation with respect to $x$, and

$$\lambda(x) = 1 - xq(x) - \int_x^1 dy \, q(y) \, . \tag{25}$$

Note that $\dot{\lambda}(x) = -x\dot{q}(x)$, and therefore when $q(x)$ is constant also $\lambda(x)$ is constant; furthermore, $\lambda(0) = \lambda(x_m) = 1 - \int_0^1 dx q(x)$, while $\lambda(1) = \lambda(x_M) = 1 - q_M$. The second term can be written as [29]

$$\lim_{n \to 0} \frac{1}{n} \log e^{\frac{1}{2} \sum_{ab} Q_{ab} \frac{\partial^2}{\partial h_a \partial h_b}} \prod_a e^{-\beta v(h_a)} \Big|_{h_a = -\sigma} = \gamma_{q_m} \star f(0, h)|_{h = -\sigma}, \tag{26}$$

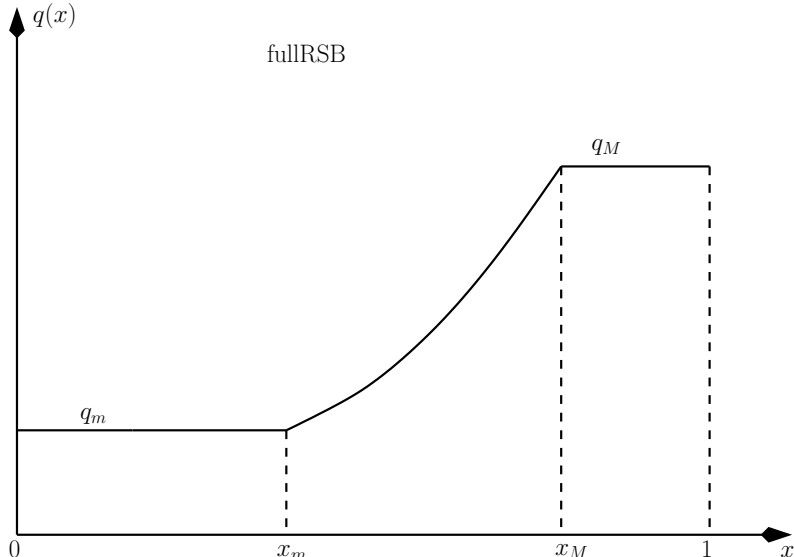

Figure 1: The expected form of the function $q(x)$ in the fullRSB phase.

where we have defined the convolution

$$\gamma_q \star g(h) = \int_{-\infty}^{\infty} \frac{\mathrm{d}y}{\sqrt{2\pi q}} e^{-\frac{y^2}{2q}} g(h-y). \tag{27}$$

The function $f(x,h)$ verifies the Parisi equation (the dot represents a $x$-differentiation, while the prime a $h$-differentiation):

$$\dot{f}(x,h) = -\frac{1}{2}\dot{q}(x)\left[f''(x,h) + xf'(x,h)^2\right], \quad x_m < x < x_M, \tag{28}$$

while for $x \notin [x_m, x_M]$ one has $\dot{q}(x) = 0$ and $\dot{f}(x,h) = 0$, hence $f(x,h)$ is independent of $x$. The boundary condition for $f(x,h)$ at $x = 1$ (or, equivalently, at $x = x_M$) is:

$$f(1,h) = \log\gamma_{1-q_M} \star e^{-\beta v(h)}. \tag{29}$$

It is useful to introduce the inverse function of $q(x)$, called $x(q)$, which is defined for $q \in [q_m, q_M]$ where $q_m$ and $q_M$ are defined in Fig. 1. Defining $f(q,h) \equiv f(x(q),h)$, Eqs. (28,29) become

$$\begin{cases} f(q_M,h) = \log\gamma_{1-q_M} \star e^{-\beta v(h)} \\ \dot{f}(q,h) = -\frac{1}{2}\left[f''(q,h) + x(q)f'(q,h)^2\right], \quad q_m < q < q_M. \end{cases} \tag{30}$$

where now the dot represents $q$-differentiation. Thus, for a given function $x(q)$ and $q_m$ and $q_M$ we can solve Eq. (30) to compute $f(q,h)$. Then the replicated free energy computed on this particular function is

$$-\beta\mathfrak{f}[x(q)] \equiv \lim_{n \to 0} \partial_n S(Q) = \frac{1}{2}\left[\log(1-q_M) + \frac{q_m}{\lambda(q_m)} + \int_{q_m}^{q_M} \frac{\mathrm{d}q}{\lambda(q)}\right] + \alpha\gamma_{q_m} \star f(q_m,h)|_{h=-\sigma}, \tag{31}$$

where we have introduced $\lambda(q) = \lambda(x(q))$, given by

$$\lambda(q) = 1 - q_M + \int_q^{q_M} \mathrm{d}p\, x(p). \tag{32}$$

The next step is to find the variational equations for the function $x(q)$, or equivalently $q(x)$.

### 3.3 Variational equations

In order to obtain the equations that determine the function $x(q)$ we need to impose that the function $f(q,h)$ that appears in Eq. (31) satisfies Eq. (30). A simple way to impose Eq. (30) is to add a Lagrange multiplier $P(q,h)$ to $s[x(q)]$. The new variational free energy thus becomes

$$
\begin{aligned}
-\beta \mathrm{f}[x(q)] = \frac{1}{2} & \left[ \log(1-q_M) + \frac{q_m}{\lambda(q_m)} + \int_{q_m}^{q_M} \frac{\mathrm{d}q}{\lambda(q)} \right] + \alpha \gamma_{q_m} \star f(q_m,h)|_{h=-\sigma} \\
& - \alpha \int \mathrm{d}h \, P(q_M,h) \left[ f(q_M,h) - \log \gamma_{1-q_M} \star e^{-\beta v(h)} \right] \\
& + \alpha \int \mathrm{d}h \int_{q_m}^{q_M} \mathrm{d}q \, P(q,h) \left\{ \dot{f}(q,h) + \frac{1}{2} \left[ f''(q,h) + x(q) f'(q,h)^2 \right] \right\} .
\end{aligned}
$$

(33)

Taking the variational equation with respect to $P(q_M,h)$ and $P(q,h)$ gives back Eq. (30). Now we can take the variational equations with respect to $f(q,h)$, $f(q_m,h)$, and $x(q)$ [20]. The resulting equations are:

$$
\begin{cases}
P(q_m,h) = \gamma_{q_m}(h+\sigma) \\
\dot{P}(q,h) = \frac{1}{2} \left[ P''(q,h) - 2x(q)(P(q,h)f'(q,h))' \right] , \quad q_m < q < q_M ,
\end{cases}
$$

(34)

and

$$
\frac{q_m}{\lambda(q_m)^2} + \int_{q_m}^{q} \mathrm{d}p \frac{1}{\lambda(p)^2} = \alpha \int \mathrm{d}h P(q,h) f'(q,h)^2 .
$$

(35)

Note that $P(q,h)$ is normalised to 1 for all $q$. Whenever $x(q)$ has a continuous part (i.e. $\dot{x}(q) \neq 0$), one can differentiate Eq. (35) w.r.t. $q$; the first and second derivatives lead, respectively, to explicit expressions of $\lambda(q)$ and $x(q)$ as functions of $f(q,h)$ and $P(q,h)$:

$$
\frac{1}{\lambda(q)^2} = \alpha \int \mathrm{d}h P(q,h) f''(q,h)^2 ,
$$

(36)

and

$$
x(q) = \frac{\lambda(q)}{2} \frac{\int \mathrm{d}h P(q,h) f'''(q,h)^2}{\int \mathrm{d}h P(q,h)[f''(q,h)^2 + \lambda(q)f''(q,h)^3]} .
$$

(37)

We stress once again that Eqs. (36) and (37) only hold whenever $\dot{x}(q) \neq 0$.

### 3.4 Iterative solution of the saddle point equations

The thermodynamic value of the free-energy at given values of the parameters $(\sigma,\alpha)$ can be obtained from the numerical solution of the variational equations. This can be obtained by iteration according to the following procedure:

1. Start with a guess for $x(q)$, $q_m \leq q \leq q_M$;

2. Use Eqs. (30) and (34) to obtain an estimate of $f(q,h)$ and $P(q,h)$;

3. Get $q_m$ from the ratio of Eqs. (35) and (36), computed in $q = q_m$;

4. Obtain a new guess for $q_M$ from Eqs. (36) and (32) computed in $q = q_M$;

5. Use Eq. (32) to obtain $\lambda(q)$ and Eq. (37) to obtain $x(q)$;

6. Iterate from 2 until convergence.

The numerical solution of the partial differential equations (30) and (34) can be obtained by discretizing the profile $x(q)$ through a stepwise function with $K$ steps (this is known as the $K$RSB approximation) and then increasing $K$ until convergence. The details of this procedure are described in Appendix B.

## 3.5 Distribution of gaps and contacts

From the general relation in Eq. (8), and using Eq. (33) we get

$$\rho(h) = \frac{1}{\alpha}\frac{\mathrm{df}}{\mathrm{d}v(h)} = e^{-\beta v(h)}\int \mathrm{d}z P(q_M, z)e^{-f(q_M, z)}\gamma_{1-q_M}(z-h) \,. \tag{38}$$

Note that $\rho(h)$ is correctly normalized so that $\int \mathrm{d}h\rho(h) = 1$. Note also that for an observable that is a function of the gaps, we have from Eq. (10):

$$[\mathcal{O}] = \int \mathrm{d}h\rho(h)\mathcal{O}(h)\theta(-h) = \int \mathrm{d}h P(q_M, h)\frac{\gamma_{1-q_M}\star[e^{-\beta v(h)}\mathcal{O}(h)\theta(-h)]}{\gamma_{1-q_M}\star e^{-\beta v(h)}} \,. \tag{39}$$

In particular the fraction of contacts is given by

$$z = \int \mathrm{d}h\,\rho(h)\theta(-h) = \int \mathrm{d}h\,P(q_M, h)\frac{\gamma_{1-q_M}\star[e^{-\beta v(h)}\theta(-h)]}{\gamma_{1-q_M}\star e^{-\beta v(h)}} \,. \tag{40}$$

# 4 The zero temperature phase diagram

The formulae derived in Sec. 3 hold for any value of the control parameters: the density of constraints $\alpha$, the parameter $\sigma$ that enters into the constraints, and the inverse temperature $\beta$. As a function of these control parameters, the order parameter function $q(x)$ has different forms, giving rise to several distinct phases and phase transitions. To simplify the study of this phase diagram, here we specialise to the zero temperature limit where a sharp SAT-UNSAT (or, equivalently, jamming) phase transition is found. Note that at finite temperature there is always a finite probability of violating some constraint, and the transition is smoothed out. The zero temperature phase diagram of the perceptron has been discussed for $\sigma \geq 0$ in [15], and for $\sigma < 0$ (but $|\sigma|$ not too large) in [16]. Here, we discuss the complete phase diagram for all values of $\sigma$ and $\alpha$, with the result given in Fig. 3, and characterize all the phases and phase transitions that appear.

## 4.1 The replica symmetric ansatz

The solution of replica equations like the ones derived in Sec. 3 is usually obtained through a series of steps. One starts by the simplest possible solution, called the "replica symmetric" (RS) solution. It corresponds to a constant $q(x) = q_M$, in which case the matrix $Q_{ab} = q_M$ for all $a \neq b$. Because $\dot{q}(x) = 0$, one also has $f(q, h) = f(q_M, h) = \log\gamma_{1-q_M}\star e^{-\beta v(h)}$ and $P(q, h) = P(q_M, h) = \gamma_{q_M}(h+\sigma)$. Thus, at the RS level we have

$$-\beta f_{RS}(q_M) = \frac{1}{2}\left[\log(1-q_M) + \frac{q_M}{1-q_M}\right] + \alpha\gamma_{q_M}\star f(q_M, h)|_{h=-\sigma} \,. \tag{41}$$

When we take the zero temperature limit of this expression, we need to specify whether we are in a SAT or UNSAT phase. We should note that the RS ansatz amounts to assume that the space of solutions form a unique connected component, or equivalently that the free energy has a single minimum [4, 20]. In the SAT phase the value of $q_M$ remains finite for $T \to 0$: because $q_M$ measures the similarity between two solutions, when the volume of the space of solutions is finite, two typical solutions differ and thus $q_M < 1$. Instead, in the UNSAT phase, under the assumption that there is a unique minimum, all the replicas converge towards the same state when $T \to 0$ and therefore one has $q \to 1$ in that limit.

### 4.1.1 The SAT phase

Taking the zero temperature limit with constant $q_M < 1$, we get

$$
\begin{aligned}
f(q_M, h) &= \lim_{\beta \to \infty} \log \gamma_{1-q_M} \star e^{-\beta v(h)} = \log \gamma_{1-q_M} \star \theta(h) \\
&= \log \int_{-\infty}^{h} \frac{\mathrm{d}z}{\sqrt{2\pi(1-q_M)}} e^{-\frac{z^2}{2(1-q_M)}} \equiv \log \Theta\left(\frac{h}{\sqrt{2(1-q_M)}}\right) \equiv f_{\text{SAT}}(q_M, h),
\end{aligned}
\tag{42}
$$

where

$$
\Theta(x) = \frac{1}{2}(1 + \operatorname{erf}(x)).
\tag{43}
$$

Note that in this case the function $-\beta f_{RS}(q_M)$ in Eq. (41) has a finite limit for $T \to 0$, which gives the entropy of the system, $s(q_M) = \lim_{T\to 0}[-\beta f_{RS}(q_M)]$, i.e. the logarithm of the volume of the space of solutions.

The saddle point equation for $q_M$ can be obtained either by taking the derivative of Eq. (41) with respect to $q_M$ or by considering explicitly the RS ansatz in Eq. (35). In the last case we obtain

$$
\frac{q_M}{(1-q_M)^2} = \alpha \int_{-\infty}^{\infty} \frac{\mathrm{d}h}{\sqrt{2\pi q_M}} e^{-\frac{(\sigma+h)^2}{2q_M}} \left[ \frac{\mathrm{d}}{\mathrm{d}h} \log \Theta\left(\frac{h}{\sqrt{2(1-q_M)}}\right) \right]^2.
\tag{44}
$$

One can solve Eq. (44) numerically, and in the SAT phase it is found, as expected, that $q_M < 1$. Within this solution, the SAT-UNSAT transition point is reached when $q_M \to 1$. Taking this limit in Eq. (44), using the asymptotic properties of the error function, we get the equation for the critical satisfiability threshold $\alpha_J(\sigma)$ within the replica symmetric ansatz:

$$
\alpha_J(\sigma) = \left[ \int_{-\infty}^{\sigma} \frac{\mathrm{d}h}{\sqrt{2\pi}} e^{-\frac{h^2}{2}} (h-\sigma)^2 \right]^{-1}.
\tag{45}
$$

For $\sigma > 0$, our optimization problem is convex, the RS solution is always stable (see Sec. 4.1.3), and Eq. (45) gives the correct result for the SAT-UNSAT transition line [15], as shown in Fig. 3.

### 4.1.2 The UNSAT phase

Within the RS ansatz, in the UNSAT phase, there is a unique energy minimum and the free energy converges to its energy. Correspondingly, $q_M \to 1$. At very low temperature, the system performs harmonic vibrations around that minimum, and in that case one can show that $q_M = 1 - \chi T + O(T^2)$. We thus take the $T \to 0$ limit with $1 - q_M = \chi T \to 0$ at the same time. Plugging this scaling in $f(q_M, h) = \log \gamma_{1-q_M} \star e^{-\beta v(h)}$, with $v(h) = h^2 \theta(-h)/2$, we get at leading order in $\beta$ (see Appendix C)

$$
f(1 - \chi/\beta, h) \simeq -\frac{\beta h^2}{2(1+\chi)} \theta(-h),
\tag{46}
$$

Therefore, the ground state energy is

$$
e_{RS} = \lim_{T \to 0} f_{RS}(q_M = 1 - \chi T) = -\frac{1}{2\chi} + \frac{\alpha}{2(1+\chi)} \int_{-\infty}^{\sigma} \frac{\mathrm{d}h}{\sqrt{2\pi}} e^{-\frac{h^2}{2}} (h-\sigma)^2.
\tag{47}
$$

and the saddle point equation for $\chi$ is given by

$$
\frac{1}{\chi^2} = \frac{\alpha}{(1+\chi)^2} \int_{-\infty}^{\sigma} \frac{\mathrm{d}h}{\sqrt{2\pi}} e^{-\frac{h^2}{2}} (h-\sigma)^2 \qquad \Leftrightarrow \qquad \left(1 + \frac{1}{\chi}\right)^2 = \frac{\alpha}{\alpha_J(\sigma)},
\tag{48}
$$

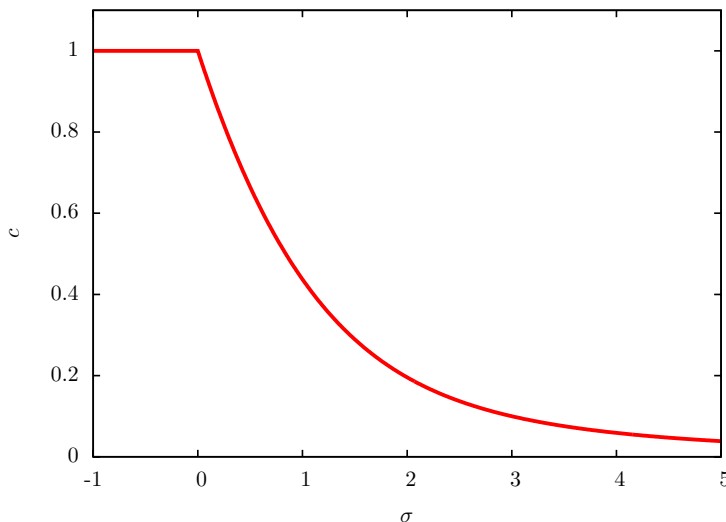

Figure 2: Isostaticity index $c$ on the jamming line, computed on the RS solution for $\sigma \geq 0$, and on the fullRSB solution for $\sigma < 0$. The system is hypostatic for $\sigma > 0$ and isostatic for $\sigma \leq 0$.

where $\alpha_J(\sigma)$ is given in Eq. (45). Eq. (48) has a solution for $\chi$ only for $\alpha > \alpha_J(\sigma)$, which indeed defines the UNSAT phase. In the SAT phase, $q_M$ remains less than one for $T \to 0$, while in the UNSAT phase we have $q_M = 1 - \chi T$. To match the two scalings, when $\alpha \to \alpha_J(\sigma)$ from the UNSAT phase, we should have that $\chi \to \infty$, which indeed follows from Eq. (48). Furthermore, inserting the saddle point equation (48) in the expression of the energy Eq. (47), we obtain

$$e_{RS} = \frac{1}{2\chi^2} = \frac{1}{2}\left(\sqrt{\frac{\alpha}{\alpha_J(\sigma)}} - 1\right)^2 . \tag{49}$$

For fixed $\sigma$ and $\alpha = \alpha_J(\sigma) + \delta\alpha$, Eq. (48) shows that $\chi^{-1} \sim \delta\alpha$ when $\delta\alpha \to 0$, and therefore, close to the jamming line, the energy goes as $e \sim \delta\alpha^2$. A similar result is obtained at fixed $\alpha$ as a function of $\delta\sigma$: the energy always scales as the square of the distance from the jamming line.

Plugging the RS ansatz in Eq. (40), and using the asymptotic scaling of $f(q_M, h)$ in Eq. (46), one gets the fraction of contacts in the UNSAT phase (at the replica symmetric level):

$$z = \int_{-\infty}^{0} \frac{\mathrm{d}h}{\sqrt{2\pi}} e^{-\frac{(h+\sigma)^2}{2}} = \Theta\left(\frac{\sigma}{\sqrt{2}}\right) . \tag{50}$$

The isostaticity condition is $c = \alpha z = 1$, and the isostaticity index $c$ is reported in Fig. 2 along the jamming line. For $\sigma > 0$ the system is hypostatic on the jamming line, while it becomes isostatic at $\sigma = 0$ [16].

### 4.1.3 Stability of the RS solution

After the RS solution, and the associated phase diagram, has been discussed, one should investigate the stability of this solution towards replica symmetry breaking (RSB). RSB can be associated to a continuous de Almeida-Thouless (dAT) instability [20], or to the discontinuous appearance of a RSB solution, usually called a "Random First Order Transition" (RFOT) [30]. In this section we concentrate on the first mechanism, which is the relevant one for the transition in the UNSAT region and in the SAT region at moderately negative values of $\sigma$.

The dAT continuous instability of the RS solution can be discussed by computing the eigenvalues of the Hessian matrix, defined as [20]

$$H_{ab;cd} = \frac{\mathrm{d}^2 S[Q]}{\mathrm{d}q_{ab}\mathrm{d}q_{cd}}\bigg|_{q_{a\neq b}=q_M} \tag{51}$$

on the RS saddle point solution $q_{a\neq b} = q_M$, where $q_M$ satisfies the RS saddle point equation. The continuous breaking of replica symmetry is associated to the vanishing of an eigenvalue of the Hessian matrix.

Here, instead of computing explicitly the eigenvalues of $H_{ab;cd}$, we follow an equivalent procedure that makes use of the fullRSB equations derived in Sec. 3. Replica symmetry breaking means that the function $q(x)$ is not a constant; continuous RSB means that $q(x)$ becomes non-constant in a continuous way, and therefore close to the instability $q(x)$ is very close to a constant. Usually, the deviation from a constant is localized around a particular point $x$. We thus assume that there is a single value of $x$ where $\dot{q}(x)$ is continuously becoming different from zero. Slightly in the unstable phase, around this point $x$, Eq. (36) holds. Upon approaching the instability point from the unstable phase, the function $q(x)$ tends to a constant and Eq. (36) reduces to its expression computed on the RS solution:

$$\frac{1}{(1-q_M)^2} = \alpha \int_{-\infty}^{\infty} \mathrm{d}h \gamma_{q_M}(h+\sigma)\left[\frac{\mathrm{d}^2}{\mathrm{d}h^2}\ln\gamma_{1-q_M}\star e^{-\beta v(h)}\right]^2 . \tag{52}$$

This expression, computed on the value of $q_M$ that satisfies the RS saddle point equation, gives a condition on $(\alpha,\sigma)$ that must be satisfied at the dAT instability, hence defining the instability line $\alpha_{\mathrm{dAT}}(\sigma)$ in the phase diagram. Note that computing Eq. (37) on this line gives the "breaking point", i.e. the value of $x$ at which the instability occurs.

In the SAT phase ($T \to 0$ at finite $q_M$), Eq. (52) reduces to

$$\frac{1}{(1-q_M)^2} = \alpha \int_{-\infty}^{\infty} \mathrm{d}h \gamma_{q_M}(h+\sigma)\left[\frac{\mathrm{d}^2}{\mathrm{d}h^2}\ln\Theta\left(\frac{h}{\sqrt{2(1-q_M)}}\right)\right]^2 . \tag{53}$$

In the UNSAT phase ($T \to 0$ with $q_M = 1 - \chi T$), it instead becomes

$$\frac{1}{\chi^2} = \frac{\alpha}{(1+\chi)^2}\int_{-\infty}^{\sigma}\frac{\mathrm{d}z}{\sqrt{2\pi}}e^{-z^2/2} , \tag{54}$$

and using Eq. (48) that gives the value of $\chi$, we get that the transition line corresponds to $\sigma = 0$, $\forall \alpha > \alpha_J(\sigma = 0) = 2$. Thus, there are two dAT transition lines where replica symmetry spontaneously breaks, one in the SAT and one in the UNSAT phase; they are reported in Fig. 3. An important conclusion of this study, which is consistent with Ref. [15], is that in the whole region $\sigma > 0$ the RS solution is stable; in particular, in that region the jamming transition happens in the RS phase and the system is hypostatic at jamming.

## 4.2 The nature of the RSB phase

We have seen in Sec. 4.1.3 that replica symmetry must be spontaneously broken in the region delimited by the dAT instability, reported in the phase diagram of Fig. 3. We now characterize the nature of the RSB transition and of the broken symmetry phase. We follow a recipe based on experience with this kind of transitions, along the following logical steps:

1. The first step is to determine the point at which the RS solution is unstable; this is the dAT line $\alpha_{\mathrm{dAT}}(\sigma)$ determined in Sec. 4.1.3. At the dAT line, the function $q(x)$ is not constant anymore. As mentioned in Sec. 4.1.3, the breaking point $m$ of the RSB solution, i.e. the point $x = m$ where $\dot{q}(x)$ becomes different from zero, can be computed plugging the RS ansatz into Eq. (37).

2. The next step is to evaluate whether $m < 1$ or $m > 1$. In fact, because the function $q(x)$ is defined for $x \in [0, 1]$, a consistent RSB solution requires $m < 1$. If this is not the case, then the dAT line cannot be a transition line, it must be preceded by a discontinuous transition of the RFOT kind. In fact one can show that the case $m = 1$ separates the two regimes. When $m = 1$, the dAT instability splits into two RFOT-like transition lines: a "dynamical transition" where a 1RSB solution appears discontinuously but the free energy remains analytical, and a Kauzmann (or condensation) transition which corresponds to a true phase transition to a spin glass phase [4, 30]. We will discuss further this situation in Sec. 4.3.

3. If $m < 1$, the dAT instability corresponds to a true continuous phase transition between a "paramagnetic" RS and a "spin glass" RSB phase. However, the dAT instability can give rise either to a fullRSB phase, with a continuous $q(x)$ for $x \in [x_m, x_M]$, or to a 1RSB phase, with $x_m = x_M = m$ and $q(x) = q_m$ for $0 < x < m$ and $q(x) = q_M$ for $m < x < 1$. To determine which one is the case, the final step is to investigate the value of $\dot{q}(m)$. In fact, a fullRSB solution requires $\dot{q}(m) > 0$, because $q(x)$ must be an increasing function[1] of $x$. If this is not the case, then the transition is a continuous transition to a 1RSB solution. To compute $\dot{q}(m)$, we derive Eq. (37), expressed as a function of $x$, with respect to $x$. We get

$$\dot{q}(x) = \left\{ \lambda^3(x) \left[ \frac{\alpha}{2} \int dh P(x, h) A(x, h) - \frac{3x^2}{\lambda^4(x)} \right] \right\}^{-1},$$
$$A(x, h) = f''''(x, h)^2 - 12x f''(x, h) f'''(x, h)^2 + 6x^2 f''(x, h)^4.$$

(55)

By evaluating Eq. (55) on the RS solution, with $x = m$, we obtain the desired result.

These steps can be performed for both dAT instabilities, in the SAT and UNSAT phases, leading to a full characterization of the RSB transition.

### 4.2.1 The SAT phase

In the SAT phase, the breaking point is

$$m = \frac{1 - q_M}{2} \frac{\int dh \gamma_{q_M}(h + \sigma) f_{\text{SAT}}'''(q_M, h)^2}{\int dh \gamma_{q_M}(h + \sigma) f_{\text{SAT}}''(q_M, h)^2 \left[ 1 + (1 - q_M) f_{\text{SAT}}''(q_M, h) \right]},$$

(56)

where $f_{\text{SAT}}(q_M, h)$ is defined in Eq. (42), and $q_M$ is the solution of the saddle point equation (44) computed at $\alpha = \alpha_{\text{dAT}}(\sigma)$. A numerical evaluation of $m$ as a function of $\sigma < 0$, on the dAT line in the SAT phase, shows that close enough to $\sigma = 0$ one has $m < 1$, while $m$ increases upon decreasing $\sigma$ towards more negative values. At $\sigma = \sigma_{\text{RFOT}}$ one has $m = 1$. Beyond that point, for $\sigma < \sigma_{\text{RFOT}}$, one has $m > 1$ and the transition becomes discontinuous, in the RFOT universality class. In order to compute the properties of the phase diagram for $\sigma < \sigma_{\text{RFOT}}$ we need to study the 1RSB solution more carefully, see Sec. 4.3.

Furthermore, the analysis of $\dot{q}(m)$ on the same transition line shows that close enough to $\sigma = 0$, for $\sigma > \sigma_{\text{1RSB}}$, one has $\dot{q}(m) > 0$ and the dAT instability leads to a fullRSB phase, while for $\sigma \in [\sigma_{\text{RFOT}}, \sigma_{\text{1RSB}}]$ one has $\dot{q}(m) < 0$ and the instability leads to a continuous transition from a RS phase towards a 1RSB phase. These results are shown in the phase diagram of Fig. 3.

---

[1]See [31] for an attempt of interpreting decreasing solutions in the replica formalism.

### 4.2.2 The UNSAT phase

Defining

$$\mathscr{F}(x) \equiv \ln \int_{-\infty}^{\infty} \frac{dz}{\sqrt{2\pi\chi}} \exp\left[ -\frac{(x-z)^2}{2\chi} - \frac{1}{2}z^2\theta(-z) \right] , \tag{57}$$

in the zero temperature limit in the UNSAT phase the breaking point is

$$m \simeq \sqrt{T}\chi(1+\chi)^3 \int \frac{dx}{\sqrt{2\pi}} \mathscr{F}'''(x)^2 \equiv \hat{m}\sqrt{T} . \tag{58}$$

The breaking point thus tends to zero proportionally to $\sqrt{T}$, and therefore in the UNSAT phase the dAT instability always leads to a consistent continuous phase transition (a discontinuous transition is never present in this case). We remark that the scaling of the breaking point along the dAT line, $m \propto \sqrt{T}$, is different from the one observed in the Sherrington-Kirkpatrick model in the zero temperature limit, where instead the breaking point remains finite. The origin of this difference will be clarified in Sec. 5.

Moreover, the zero temperature limit of the slope $\dot{q}(m)$ at the breaking point, in the UNSAT phase, is given by

$$\dot{q}(m) \simeq \left[ \frac{\alpha\chi^3}{2} \frac{1}{\sqrt{T}} \int \frac{dx}{\sqrt{2\pi}} \mathscr{F}''''(x)^2 \right]^{-1} \geq 0 , \tag{59}$$

which is of order $\sqrt{T}$ and always positive. This implies that the dAT instability line in the UNSAT phase is a transition from a RS solution to a fullRSB one [16], see Fig. 3. We will see in Sec. 5 that the scaling with $T$ of both $m$ and $\dot{q}(m)$ is in perfect agreement with the complete scaling form of $q(x)$ in the UNSAT phase.

### 4.3 The 1RSB free-energy in the SAT phase

In Sec. 4.2 we have shown that for $\sigma < \sigma_{\text{RFOT}}$ the solution for $q(x)$ in the SAT phase becomes of the RFOT type (i.e. a discontinuous 1RSB solution) [30]. In order to characterize this part of the phase diagram we need to consider explicitly a general 1RSB ansatz. The 1RSB free energy is characterized by three numbers $q_0 = q_m$, $q_1 = q_M$ and $m$, the function $q(x)$ being equal to $q_m$ for $x \in [0, m]$ and to $q_M$ for $x \in [m, 1]$ (see Appendix B for a general $K$RSB solution). At zero temperature, in the SAT phase where both $q_0 < 1$ and $q_1 < 1$, the function $-\beta f[x(q)]$ has a finite limit, which coincides with the 1RSB entropy, given by

$$s_{1RSB}(q_1, q_0, m) = \frac{m-1}{2m} \log(1-q_1) + \frac{1}{2m} \log(1 + m(q_1 - q_0) - q_1) + \frac{q_0}{2(1 + m(q_1 - q_0) - q_1)}$$

$$+ \frac{\alpha}{m} \int \frac{dt}{\sqrt{2\pi q_0}} e^{-\frac{t^2}{2q_0}} \log\left[ \gamma_{q_1-q_0} \star \Theta^m\left( \frac{t-\sigma}{\sqrt{2(1-q_1)}} \right) \right] . \tag{60}$$

The variational equations always admit the RS solution $q_1 = q_0$. For any given $\sigma$ this is the unique solution for sufficiently small $\alpha$. For fixed $\sigma < \sigma_{\text{RFOT}}$ and upon increasing $\alpha$, there is a point, called the dynamical transition $\alpha_{\text{dyn}}(\sigma)$, where a non-trivial solution with $q_1 \neq q_0$ appears for $m \to 1$. We do not discuss in details the properties of this transition, for which we refer to [4, 30, 32]. In short, at $\alpha_{\text{dyn}}(\sigma)$ the Boltzmann-Gibbs measure on the space of solutions splits into an exponential number of clusters of solutions [4]. These clusters can be classified according to their internal entropy that measures their typical size. At the dynamical transition point, the clusters that dominate the Boltzmann-Gibbs measure are the most numerous ones, meaning the ones with higher complexity (or configurational entropy). It is important to note that the free energy remains analytic upon crossing $\alpha_{\text{dyn}}(\sigma)$, which is thus not a thermodynamic phase transition [4,30]. Increasing $\alpha$, the complexity of the relevant

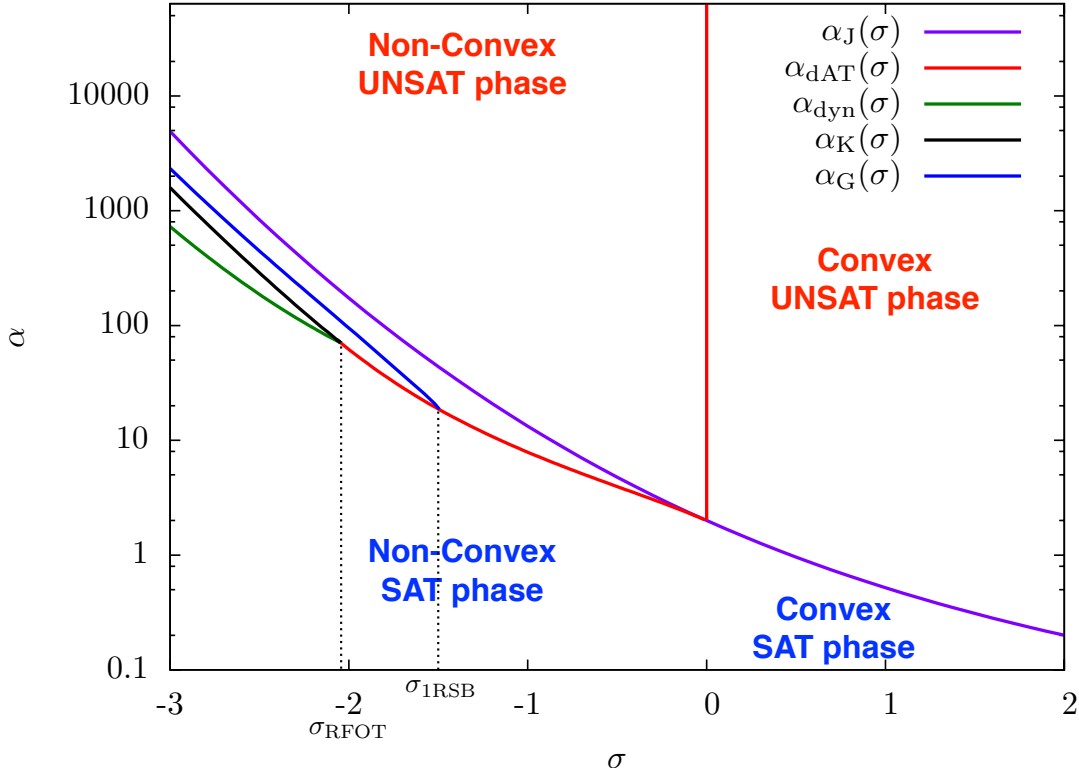

Figure 3: The zero temperature phase diagram of the perceptron. $\alpha_J(\sigma)$ is the jamming transition (or SAT/UNSAT threshold) within the replica symmetric approximation. The replica symmetric solution is stable for all $\sigma > 0$. Thus the jamming transition line is exact only for $\sigma > 0$. $\alpha_{dAT}(\sigma)$ are the lines where replica symmetry breaks. In the non-convex region and for $\sigma \in [\sigma_{1RSB}, 0]$, $\alpha_{dAT}$ is a transition from a replica symmetric phase to a continuous fullRSB one. For $\sigma \in [\sigma_{RFOT}, \sigma_{1RSB}]$ the dAT line is a continuous transition from a replica symmetric phase to a stable 1RSB phase. For $\sigma < \sigma_{RFOT}$ a RFOT type phenomenology is observed. Keeping $\sigma$ fixed and increasing $\alpha$, one has first a dynamical transition for $\alpha = \alpha_{dyn}(\sigma)$, then a Kauzmann transition for $\alpha = \alpha_K(\sigma)$, and finally a Gardner transition for $\alpha = \alpha_G(\sigma)$. In the UNSAT phase, the instability line $\alpha_{dAT}(\sigma)$ represents the transition from a replica symmetric phase to a continuous fullRSB one for all $\alpha > 2$. Therefore, fullRSB occurs in the region delimited by the $\alpha_{dAT}(\sigma)$ and $\alpha_G(\sigma)$ lines, which contains the whole jamming line for $\sigma < 0$.

clusters decreases and vanishes at the Kauzmann (or condensation) transition $\alpha = \alpha_K(\sigma)$ [4, 30, 32]. At this point, the clusters that dominate the Boltzmann-Gibbs measure are no longer exponentially many, and the system undergoes a thermodynamic phase transition. Note that this two-transition scenario, also called RFOT, is at the basis of the mean field theory of glasses (see [27, 33–36] for reviews).

To compute $\alpha_{dyn}(\sigma)$ and $\alpha_K(\sigma)$, we are therefore led to consider the 1RSB entropy for $m \approx 1$, where we can write to the leading order

$$s_{1RSB}(q_1, q_0, m) = s_{RS}(q_0) + (m-1)\delta s_{RSB}(q_0, q_1),$$  (61)

with $s_{RS}$ given in Eq. (41), and

$$\delta s_{RSB}(q_0, q_1) = -s_{RS}(q_0) - \frac{q_0^2}{2(1-q_0)^2} + \frac{q_1(1-2q_0)}{2(1-q_0)^2} + \frac{1}{2}\log(1-q_1) + \alpha \int \frac{dt}{\sqrt{2\pi q_0}} \, e^{-\frac{t^2}{2q_0}} \times$$

$$\int \frac{du}{\sqrt{2\pi(q_1-q_0)}} \, e^{-\frac{u^2}{2(q_1-q_0)}} \, \frac{\Theta\left(\frac{u+t-\sigma}{\sqrt{2(1-q_1)}}\right)}{\Theta\left(\frac{t-\sigma}{\sqrt{2(1-q_0)}}\right)} \log\Theta\left(\frac{u+t-\sigma}{\sqrt{2(1-q_1)}}\right). \qquad (62)$$

When $m \to 1$, $q_0$ can be obtained by setting to zero the derivative of Eq. (41), which gives the saddle point equation (44). The dynamical transition point $\alpha_{\rm dyn}(\sigma)$ corresponds to the lowest value of $\alpha$ where the variational equation for $q_1$, obtained by setting the derivative of Eq. (62) to zero, admits a solution $q_1 > q_0$. At the Kauzmann transition point $\alpha_{\rm K}(\sigma)$, the breaking point $m$, obtained by setting the derivative with respect to $m$ of Eq. (60) to zero, first becomes smaller than one. Upon further increasing $\alpha$, it can be shown that the 1RSB solution becomes unstable and undergoes a Gardner transition towards a fullRSB phase [37]. The equation that controls the Gardner instability of the 1RSB solution can also be obtained using the fullRSB equations, similarly to what has been done in Sec. 4.1.3 for the RS solution. In the following we assume that the continuous part of $q(x)$ appears in correspondence to the value $q_1$, as it usually happens at a Gardner transition [37] (the stability towards a continuous breaking at $q_0$ can be discussed along similar lines). If $q_1$, $q_0$ and $m$ satisfy the saddle point conditions, the instability point $\alpha_{\rm G}(\sigma)$ of the 1RSB transition is given by the following condition:

$$\frac{1}{\lambda_{1RSB}^2(1)} = \alpha \int dh \, P_{1RSB}(1,h) f_{1RSB}''(1,h)^2 \,, \qquad (63)$$

where

$$\lambda_{1RSB}(1) = 1 - q_1 \,,$$
$$f_{1RSB}(1,h) = \log\Theta\left[\frac{h}{\sqrt{2(1-q_1)}}\right] \,,$$
$$P_{1RSB}(1,h) = \Theta^m\left[\frac{h}{\sqrt{2(1-q_1)}}\right] \gamma_{q_1-q_0} \star \left[\gamma_{q_0}(h+\sigma) e^{-m f_{1RSB}(m,h)}\right] \,, \qquad (64)$$
$$f_{1RSB}(m,h) = \frac{1}{m}\log\left[\gamma_{q_1-q_0} \star \Theta^m\left[\frac{h}{\sqrt{2(1-q_1)}}\right]\right] \,.$$

The dynamical, the Kauzmann and Gardner transition lines are plotted in Fig. 3.

## 5 The SAT-UNSAT transition and its critical properties

Having established the phase diagram of the zero temperature random perceptron model (Fig. 3), we discuss here the critical properties of the jamming (or SAT-UNSAT) transition line. The jamming line at $\sigma > 0$ has been discussed already in [14, 15] and it is known to be non-critical, in the sense that the system is hypostatic (see Fig. 2), which according to the analysis of [7] does not lead to marginal stability. In this section we want to describe the jamming transition for $\sigma < 0$, which falls in the fullRSB phase (Fig. 3): in this case, the system is isostatic at jamming and a non-trivial critical behavior appears [14]. The jamming point can be approached both from the SAT and UNSAT phase. From the SAT (unjammed) phase, upon approaching the transition the volume of the space of solutions shrinks to zero and the

self-overlap in a cluster of solutions is asymptotically close to one. From the UNSAT (jammed) phase, the energy goes to zero upon approaching the transition. In both cases, universality naturally emerges with a set of nontrivial critical exponents that characterize the scaling of physical quantities on both sides of the transition. We did not attempt to solve numerically the fullRSB equations (see [13] for a numerical study of the equations in the hard sphere case); in fact, the values of the critical exponents can be extracted analytically via a scaling analysis of the equations, that we discuss in the rest of this section.

## 5.1 Asymptotes of $f(q,h)$ and $P(q,h)$

For later use let us discuss the asymptotic behavior of $f(q,h)$ and $P(q,h)$ for $h \to \pm\infty$ which are generically valid, in particular in the scaling solutions of our interest. For $h \to \pm\infty$ the boundary condition for $f(q,h)$ reduces to

$$f(q_M, h \to \infty) = 0 \quad \text{and} \quad f(q_M, h \to -\infty) = -\frac{h^2}{2}\frac{\beta}{1+\beta(1-q_M)}. \tag{65}$$

Using Eq. (30) one readily finds

$$f(q, h \to \infty) = 0 \,, \qquad f(q, h \to -\infty) \sim -\frac{h^2}{2}\frac{\beta}{1+\beta\lambda(q)} \,. \tag{66}$$

For $h \to \infty$, Eq. (34) becomes simply $\dot{P} = P''/2$, which has a unique solution compatible with the boundary condition

$$P(q, h \to \infty) = \gamma_q(h + \sigma) \,. \tag{67}$$

For $h \to -\infty$ the equation is slightly more complicated; but still, a Gaussian form of the kind $P(q,h) \sim \sqrt{D(q)}e^{-D(q)h^2}$ satisfies Eq. (34) for large negative $h$, where $D(q)$ verifies

$$\dot{D}(q) = -2D(q)^2 + 2D(q)\frac{\beta x(q)}{1+\beta\lambda(q)} \,,$$
$$D(q_m) = \frac{1}{2q_m} \,. \tag{68}$$

## 5.2 Approaching jamming from the SAT phase

In the SAT phase we can take the limit $T = 1/\beta \to 0$ by simply replacing $e^{-\beta v(h)} \to \theta(h)$. The volume of the space of solutions is finite and the resulting equations are well defined and give a value $q_M < 1$. This is due to the fact that zero-energy configurations are not isolated: they form clusters where two typical solutions have overlap $q_M < 1$. Solutions that belong to different clusters have overlap $q < q_M$ whose statistical properties are described by the function $x(q)$: it represents the (average) probability that two configurations have an overlap smaller than $q$ [20]. In the jamming limit, the cluster volumes go to zero and therefore the self-overlap of a cluster, $q_M$, goes to one.

### 5.2.1 Scaling form of the solution close to jamming

Close to jamming the fullRSB equations develop a scaling regime, as can be deduced from a numerical analysis [13] (see Sec. 3.4 and Appendix B for details on how to solve the fullRSB equations numerically). In the scaling regime, it is convenient to make the following change of variables:

$$y(q) = \epsilon^{-1}x(q) \,,$$
$$\widehat{f}(q,h) = \epsilon f(q,h) \,,$$
$$\widehat{\lambda}(q) = \epsilon^{-1}\lambda(q) \,, \tag{69}$$

where $\epsilon$ is the linear distance from the jamming line (either in $\alpha$ or $\sigma$). In addition we assume that $q_M = 1 - \epsilon^{\kappa}$ where $\kappa$ is an exponent to be determined by the equations. In the jamming limit $\epsilon \to 0$, one has $y \in [0, 1/\epsilon] \to [0, \infty)$ and $q \in [q_m, q_M] \to [q_m, 1]$.

We wish to show that a scaling solution exists in this limit, when $y$ is large and $q \sim 1$. It has the following form:

$$y(q) \sim y_J (1-q)^{-1/\kappa}$$

$$P(q,h) \sim \begin{cases} (1-q)^{(1-\kappa)/\kappa} p_-[h(1-q)^{(1-\kappa)/\kappa}] & \text{for } h \sim -(1-q)^{(\kappa-1)/\kappa} \\ (1-q)^{-a/\kappa} p_0\left(\frac{h}{\sqrt{1-q}}\right) & \text{for } |h| \sim \sqrt{1-q} \\ p_+(h) & \text{for } h \gg \sqrt{1-q} \end{cases} \tag{70}$$

$$m(q,h) = \widehat{\lambda}(q)\widehat{f}'(q,h) = -\sqrt{1-q}\,\mathcal{M}\left[\frac{h}{\sqrt{1-q}}\right], \quad \mathcal{M}(t \to \infty) = 0, \quad \mathcal{M}(t \to -\infty) = t.$$

While the functions $p_-$ and $p_+$ and the equations that they verify are peculiar of the perceptron model and depend on the parameters $\sigma$ and $\alpha$, on the jamming line the function $p_0$ as well as the exponents $a$ and $\kappa$ are universal (i.e. independent of the precise location on the jamming line). In Sec. 5.2.2 we show that the universal equations determining $p_0$ and the critical exponents also coincide with the ones obtained for the jamming transition of hard spheres in high dimension. Note that for finite $\epsilon$, the scaling solution (70) is cutoff when $q \sim q_M = 1 - \epsilon^{\kappa}$ and $y \sim y_M = Y\epsilon^{-1}$. Finally, note that while we will be able to prove the existence of the scaling solution, and compute the values of $a$ and $\kappa$, we will not be able to prove that $\epsilon$ is proportional to the distance from the jamming line: at present, this follows from the numerical solution of the equations [13].

### 5.2.2 Proof of the scaling form

The scaling analysis is carried out along the lines of [13, 14, 38].

**Scaling of $f(q,h)$** – The function $m(q,h)$ introduced in Eq. (70) satisfies the equation

$$\dot{m}(q,h) = -\frac{1}{2}m''(q,h) - \frac{y(q)}{\widehat{\lambda}(q)}m(q,h)[1 + m'(q,h)],$$

$$m(q, h \to -\infty) \simeq -h, \qquad m(q, h \to \infty) \simeq 0, \tag{71}$$

where the differential equation comes from Eq. (30) and the asymptotes from Eq. (66). Let us inspect the value of $\frac{y(q)}{\widehat{\lambda}(q)}$ for $\epsilon \to 0$ and $q \to 1$, according to Eq. (70):

$$\frac{y(q)}{\widehat{\lambda}(q)} = \frac{y_J(1-q)^{-1/\kappa}}{\epsilon^{\kappa-1}[1 - y_J\frac{\kappa}{\kappa-1}] + y_J\frac{\kappa}{\kappa-1}(1-q)^{1-1/\kappa}}. \tag{72}$$

This expression has a crossover for $1 - q \sim \epsilon^{\kappa}$, when the two terms in the denominator are of the same order. The scaling form is obtained for $1 - q \gg \epsilon^{\kappa}$, in which case we have

$$\frac{y(q)}{\widehat{\lambda}(q)} = \frac{\kappa - 1}{\kappa}\frac{1}{1-q}. \tag{73}$$

Plugging this result and the scaling form (70) in Eq. (5.2.2), we obtain a scaling equation for the function $\mathcal{M}(t)$:

$$\mathcal{M}(t) - t\mathcal{M}'(t) = \mathcal{M}''(t) + 2\frac{\kappa - 1}{\kappa}\mathcal{M}(t)[1 - \mathcal{M}'(t)],$$

$$\mathcal{M}(t \to \infty) = 0, \qquad \mathcal{M}(t \to -\infty) = t. \tag{74}$$

This non-linear equation, with boundary conditions at $t \to \pm\infty$, admits a unique solution for each value of $\kappa$.

**Scaling of $P(q,h)$** – The existence of the functions $p_-$ and $p_+$ and the corresponding scaling variables can be obtained by the analysis of the equation for $P$ at large negative and positive arguments, respectively. From Eq. (67) it follows that for $h \to \infty$, $P(q,h)$ remains a finite Gaussian, $P(q,h \to \infty) \sim \gamma_q(h+\sigma)$. For finite $h > 0$, the Gaussian will be deformed to a finite function $p_+(h)$ as it appears in Eq. (70). For $h \to -\infty$, on the other hand, we have from Eq. (68) that for $\beta \to \infty$ and in the scaling regime of Eq. (73):

$$\dot{D}(q) = -2D(q)^2 + 2D(q)\frac{y(q)}{\widehat{\lambda}(q)} = -2D(q)^2 + 2D(q)\frac{\kappa-1}{\kappa}\frac{1}{1-q}\ . \tag{75}$$

If $\kappa < 2$ (which, we will see, is the case), this equation admits a scaling solution

$$D(q) \sim D_J (1-q)^{-2(\kappa-1)/\kappa}, \tag{76}$$

where the term $2D(q)^2$ is negligible. One concludes therefore that in the scaling regime, $P(q,h \to -\infty) \sim \sqrt{D(q)}e^{-D(q)h^2}$ has the form

$$P(q,h) \sim (1-q)^{(1-\kappa)/\kappa}p_-[h(1-q)^{(1-\kappa)/\kappa}] \tag{77}$$

which has been proven asymptotically for $h \to -\infty$ but can be extended to the whole regime where $q \sim 1$ and $|h| \sim (1-q)^{(\kappa-1)/\kappa}$, as in Eq. (70). Note that even if it has a scaling form, the function $p_-$ is *not* uniquely determined by the scaling regime, and remains non-universal, see [38].

The "matching" regime with $p_0(t)$ must be introduced in Eq. (70) to smoothly match the two regimes for negative and positive $h$. Here is where universality appears. Matching $p_0$ and $p_+$ requires that

$$p_+(t \to 0^+) \sim t^{-\gamma}\ , \qquad p_0(t \to \infty) \sim t^{-\gamma}\ , \qquad \gamma = \frac{2a}{\kappa}\ . \tag{78}$$

Matching $p_-$ and $p_0$ requires that

$$p_-(t \to 0^-) \sim |t|^{\theta}\ , \qquad p_0(t \to -\infty) \sim |t|^{\theta}\ , \qquad \theta = \frac{1-\kappa+a}{\kappa/2-1}\ . \tag{79}$$

The scaling variable $t = h/\sqrt{1-q}$ appearing in the matching regime is naturally the same as for $f(q,h)$. This is in fact the only choice that leads, once plugged in Eq. (34), to a non-trivial equation for $p_0(t)$, namely:

$$\frac{a}{\kappa}p_0(t) + \frac{1}{2}t\, p_0'(t) = \frac{1}{2}p_0''(t) + \frac{\kappa-1}{\kappa}(p_0(t)\mathcal{M}(t))'\ ,$$
$$p_0(t \to \infty) = t^{-2a/\kappa}\ , \qquad p_0(t \to -\infty) = |t|^{(1-\kappa+a)/(\kappa/2-1)}\ . \tag{80}$$

Recall that in Eq. (5.2.2), $\mathcal{M}(t)$ depends on $\kappa$; it turns out that Eq. (5.2.2) also admits a unique solution for $p_0(t)$ satisfying the correct asymptotic conditions, *but only for a given choice of $a = a(\kappa)$*. For a given $\kappa$, Eqs. (5.2.2) and (5.2.2) thus determine $\mathcal{M}(t)$, $p_0(t)$ and $a$.

**Determination of the exponent $\kappa$** – The exponent $\kappa$ can be fixed using Eq. (37) which can be equivalently written as

$$\frac{y(q)}{\widehat{\lambda}(q)} = \frac{1}{2}\frac{\int dh P(q,h)m''(q,h)^2}{\int dh P(q,h)m'(q,h)^2[1+m'(q,h)]}\ . \tag{81}$$

In the scaling regime, Eq. (73) gives the left hand side. In the right hand side, we can note that $m''(q,h)$ and $m'(q,h)^2[1 + m'(q,h)]$ both vanish outside the scaling regime, because of the asymptotic behavior of $m(q,h)$. Therefore, the right hand side only receives contribution from the regime $h \sim \sqrt{1-q}$. We obtain

$$\frac{\kappa - 1}{\kappa} = \frac{1}{2} \frac{\int dt \, p_0(t) \mathscr{M}''(t)^2}{\int dt \, p_0(t) \mathscr{M}'(t)^2 [1 + \mathscr{M}'(t)]} \, . \tag{82}$$

Because both $\mathscr{M}(t)$ and $p_0(t)$ depend on $\kappa$, this is an equation for $\kappa$. The numerical solution of the system of Eqs. (5.2.2), (5.2.2) and (82), gives $\kappa = 1.41574\ldots$ while within the numerical precision one finds the relation [38]

$$a = 1 - \frac{\kappa}{2} \quad \Rightarrow \quad \gamma = \frac{2 - \kappa}{\kappa} \, , \quad \theta = \frac{3\kappa - 4}{2 - \kappa} \quad \text{and} \quad \gamma = \frac{1}{2 + \theta} \, . \tag{83}$$

The importance of these "scaling" relations will be further discussed in Sec. 5.5 and Sec. 5.6.

## 5.3 Zero temperature fullRSB solution in the UNSAT phase

We now turn to the analysis of the approach to the jamming transition from the UNSAT phase. Before doing that, however, we need to study the behavior of the fullRSB solution for $T \to 0$ in the UNSAT phase. In Sec. 4.2 we have shown that in the UNSAT phase, close to the dAT instability line ($\sigma = 0, \alpha > 2$) of the RS solution, the solution $q(x)$ has the following properties:

- The Edwards-Anderson order parameter is $q_M = 1 - \chi T$

- The breaking point at the instability transition line is $m = \hat{m}\sqrt{T}$

- The slope of $q(x)$ at the breaking point on the instability line is $\dot{q}(m) \sim \sqrt{T}$

Additionally, here we want to show that, like in the Sherrington-Kirkpatrick model [20,39]:

- $P(q,h)$ is smooth and regular at zero temperature

- The function $\beta x(q)$ admits a finite zero temperature limit

For small temperature in the UNSAT phase, the initial condition for $f(q_M,h)$ is given by (Appendix C):

$$f(q_M,h) = \begin{cases} -\frac{\beta h^2}{2(1+\chi)} \theta(-h) & \text{for } |h| \gg \sqrt{T} \, , \\ \mathscr{F}(h/\sqrt{T}) & \text{for } |h| \sim \sqrt{T} \, , \end{cases} \tag{84}$$

with $\mathscr{F}(x)$ defined in Eq. (57). In the fullRSB region, as in the RS case, we define $q_M = 1 - \chi T$ and we introduce

$$y(q) = \frac{\beta}{\chi} x(q) \, , \qquad \widehat{f}(q,h) = T\chi f(q,h) \, , \qquad \widehat{\lambda}(q) = \frac{\beta}{\chi} \lambda(q) = 1 + \int_q^1 dp \, y(p) \, . \tag{85}$$

As in Sec. 5.2.2, we introduce $m(q,h) = \widehat{\lambda}(q)\widehat{f}'(q,h) = \lambda(q)f'(q,h)$, which satisfies Eq. (5.2.2) with the modification $m(q, h \to -\infty) = -h\chi\widehat{\lambda}(q)/(1 + \chi\widehat{\lambda}(q))$ and initial condition

$$m(1,h) = -\frac{h\chi}{1 + \chi} \theta(-h) \, . \tag{86}$$

The equation for $P(q,h)$ is still Eq. (34). Finally, Eq. (36) becomes

$$\frac{1}{\chi^2} = \alpha \frac{\int dh P(1,h)\theta(-h)}{(1 + \chi)^2} \, , \tag{87}$$

and Eq. (37) becomes identical to Eq. (81). The distribution of gaps, given in Eq. (38), becomes

$$\rho(h) = \begin{cases} P(1, h(1+\chi))(1+\chi) & \text{for } h < 0 \, , \\ P(1, h) & \text{for } h > 0 \, . \end{cases} \tag{88}$$

Because the breaking point $m = x(q_M) \propto \sqrt{T}$, one has $y(q_M) \sim 1/\sqrt{T}$ and thus $y(q)$ extends up to infinity in the zero temperature limit. The scaling behavior of $y(q)$ for $q \to 1$ is expected to be [39]:

$$y(q) \simeq \frac{y_\chi}{\sqrt{1-q}} \, . \tag{89}$$

We now check that this scaling is consistent. First of all, this implies that $q(x) \sim 1 - AT^2/x^2$ for some constant $A$. For $x = \hat{m}\sqrt{T}$, we get $q_M = 1 - \chi T$ which matches the behavior on the instability line. Also, one has $\dot{q}(x) \propto T^2/x^3$ and then $\dot{q}(m) \propto \sqrt{T}$, as it is the case along the instability line. The final check can be obtained from Eq. (81). We assume that the scaling behavior of $m(q, h)$ for $q \to 1$ is

$$m(q, h) \sim -\frac{\chi}{1+\chi}\sqrt{1-q}\mathcal{M}\left(\frac{h}{\sqrt{1-q}}\right), \quad \mathcal{M}(t \to \infty) = 0 \, , \quad \mathcal{M}(t \to -\infty) = t \, , \tag{90}$$

which agrees with the initial condition (86). Plugging this ansatz inside Eq. (81) we get

$$y(q) = \frac{1}{2}\frac{1+\chi}{\sqrt{1-q}}\frac{P(1,0)\int_{-\infty}^{\infty} dt(\mathcal{M}''(t))^2}{\int_{-\infty}^{0} dh P(1,h)} \quad \Rightarrow \quad y_\chi = \frac{1}{2}(1+\chi)\frac{P(1,0)\int_{-\infty}^{\infty} dt(\mathcal{M}''(t))^2}{\int_{-\infty}^{0} dh P(1,h)} \, , \tag{91}$$

which confirms the consistency of Eq. (89) and provides an explicit expression of $y_\chi$. The only point left to verify is that $P(1, 0)$, and more generally $P(1, h)$ are finite. First, we note that for $h \to \infty$ we have $P(q, h) \to \gamma_q(h + \sigma)$, which suggests that for $h \to \infty$, $P(1, h)$ is finite and has smooth corrections in $q$. Next, we can observe that Eq. (68) becomes for $q \to 1$:

$$\dot{D}(q) = -2D(q)^2 + 2D(q)\left(\frac{y_\chi}{\sqrt{1-q}} + \cdots\right) \quad \Rightarrow \quad D(q) = D_0 - 4D_0 y_\chi \sqrt{1-q} + \cdots \tag{92}$$

which therefore indicates that for $h \to -\infty$, $P(1, h)$ is finite and has corrections proportional to $\sqrt{1-q}$. Plugging $P(q, h) \sim P(1, h) + \sqrt{1-q}\,\delta P(h)$ in Eq. (34), and using Eq. (90), we obtain

$$\begin{aligned} \delta P(h) &= -2y_\chi \frac{\chi}{1+\chi} \lim_{q \to 1} \frac{d}{dh}\left[P(1, h)\sqrt{1-q}\mathcal{M}\left(\frac{h}{\sqrt{1-q}}\right)\right] \\ &= -2y_\chi \frac{\chi}{1+\chi}\theta(-h)[hP'(1, h) + P(1, h)] \end{aligned} \tag{93}$$

which satisfies the condition $\int dh\, \delta P(h) = 0$ and shows that there are $\sqrt{1-q}$ corrections only in the region $h < 0$.

## 5.4 The jamming limit from the UNSAT phase

The jamming limit from the UNSAT phase is obtained by considering the $T = 0$ equations of Sec. 5.3 in the limit $\chi \to \infty$, as in the RS case. This is because $q_M$ is finite in the SAT phase when $T = 0$, while in the UNSAT phase $q_M = 1 - \chi T$ for $T \to 0$: matching the two regimes for $T \sim 0$ requires the divergence of $\chi$ at the jamming point.

We know that in the UNSAT phase for $q \to 1$ we have, from Eq. (89) and (85):

$$\frac{y(q)}{\widehat{\lambda}(q)} \sim \frac{y_\chi / \sqrt{1-q}}{1 + 2y_\chi \sqrt{1-q}} \ . \tag{94}$$

If $\chi$ is finite and $q \to 1$, then in the denominator the factor 1 dominates and $y/\widehat{\lambda} \sim (1-q)^{-1/2}$: in this regime we recover the "regular" zero temperature solution of the UNSAT phase described in Sec. 5.3. Conversely, for $\chi \to \infty$, the coefficient $y_\chi \to \infty$. In fact, from Eq. (91) we obtain $y_\chi \propto \chi P(1,0)$ observing that $\mathcal{M}(t)$ is finite and that $\int_{-\infty}^0 \mathrm{d}h P(1,h)$ must remain finite because $P(1,h)$ is normalized to 1. In this case, for all $q < 1$, at large enough $\chi$ the second term in the denominator of Eq. (94) dominates and one has

$$\frac{y(q)}{\widehat{\lambda}(q)} \sim \frac{1}{2(1-q)} \ , \qquad \text{for } q \sim 1 \text{ and } y_\chi \sqrt{1-q} \sim \chi P(1,0)\sqrt{1-q} \gg 1 \ . \tag{95}$$

In this regime, we must have a different scaling solution in which $y/\widehat{\lambda} \propto 1/(1-q)$: but this is exactly the jamming scaling solution that was already derived in Sec. 5.2, which indeed must emerge from the regular zero temperature UNSAT solution upon approaching the jamming point.

To summarize, when $\chi \to \infty$ and in the region of $q \to 1$, we expect two different scaling solutions: when $1 - q \ll 1 - q_*$, we have the "regular" UNSAT scaling of Sec. 5.3; for $1 - q \gg 1 - q_*$ we have instead the "jamming" scaling solution of Sec. 5.2. The matching point $q_*$ is determined by the condition that $\chi P(1,0)\sqrt{1-q_*} \sim 1$, and the value of $y(q_*)$ is then simply

$$y(q_*) = \frac{\chi P(1,0)}{\sqrt{1-q}} \sim \frac{1}{1-q_*} \ . \tag{96}$$

Note that in the jamming solution, Eq. (70), we have $P(q_*, 0) \sim (1-q_*)^{-a/\kappa}$, and we know from Eq. (93) that in the regular solution $P(q,h)$ changes by a very small amount, $\sim \sqrt{1-q}$, when $q \to 1$. We conclude that $P(1,0) \sim (1-q_*)^{-a/\kappa}$ and more generally

$$P(1,h) \sim (1-q_*)^{(1-\kappa)/\kappa} p_-((1-q_*)^{(1-\kappa)/\kappa} h) \ , \tag{97}$$

for negative values of $h$. We obtain therefore

$$\chi P(1,0)\sqrt{1-q_*} \sim \chi(1-q_*)^{1/2-a/\kappa} \sim 1 \quad \Rightarrow \quad \chi \sim (1-q_*)^{a/\kappa - 1/2} \quad \Rightarrow \quad 1 - q^* \sim \chi^{\frac{2\kappa}{2a-\kappa}} \ , \tag{98}$$

which concludes the analysis of the matching between the two scaling solutions on the UNSAT side of the transition.

## 5.5 Scaling relations between exponents

The relation $a = 1 - \kappa/2$, and its consequences that are given in Eq. (83), has been found in Ref. [13] within numerical precision by solving the equations for the critical exponents derived in Sec. 5.2.2. Eq. (83) is physically very important: in fact, the relation $\gamma = 1/(2+\theta)$ has been proven in [7] to be a direct consequence of marginal stability (in the perceptron, the same marginal stability argument is discussed in [14]). It would therefore be nice to have a more direct analytical proof of the relation $a = 1 - \kappa/2$, that does not rely on a numerical calculation.

While in principle it must be possible to obtain such a proof directly from the properties of the equations of Sec. 5.2.2, that define all the critical exponents, here we give an independent argument by showing how the matching condition between the two asymptotic solutions allows one to derive analytically this relation. To this aim, we focus on the pressure

$p = -[h]$. In Appendix D, we show that $p \propto 1/\chi^2$ in the jamming limit $\chi \to \infty$. Using now $P(1,h) \approx (1-q_*)^{(1-\kappa)/\kappa}p_-((1-q_*)^{(1-\kappa)/\kappa}h)$ for $h < 0$, which was derived in Sec. 5.4, together with Eq. (88), we find

$$\rho(h < 0) \sim \chi(1-q_*)^{(1-\kappa)/\kappa}p_-[\chi(1-q_*)^{(1-\kappa)/\kappa}h] \qquad \Rightarrow \qquad [h] \sim \frac{1}{\chi(1-q_*)^{(1-\kappa)/\kappa}} . \quad (99)$$

Because we know that $[h] \propto 1/\chi^2$, we must have

$$\chi \sim (1-q_*)^{(1-\kappa)/\kappa} . \quad (100)$$

This is compatible with (98) only if $a = 1-\kappa/2$, which gives an independent proof of Eq. (83). Finally, recalling that $P(q_*,h) \sim P(1,h)$, and using Eq. (100) into Eq. (70), we obtain

$$P(1,h) \sim P(q_*,h) \sim \begin{cases} \chi\, p_-(h\chi) & \text{for } h \sim -\chi^{-1} \\ \chi^{-\frac{2-\kappa}{2(1-\kappa)}} p_0\left(h\chi^{-\frac{\kappa}{2(1-\kappa)}}\right) & \text{for } |h| \sim \chi^{\frac{\kappa}{2(1-\kappa)}} \\ p_+(h) & \text{for } h \gg \chi^{\frac{\kappa}{2(1-\kappa)}} \end{cases} \quad (101)$$

which holds on the UNSAT side upon approaching jamming.

## 5.6 Scaling of several interesting observables at the jamming transition

We have now fully characterized the scaling of the basic quantities, $x(q)$, $\lambda(q)$, $f(q,h)$, and $P(q,h)$, in the vicinity of the jamming transition, both on the SAT and on the UNSAT side. On the SAT side, our main results are Eqs. (69) and (70), together with Eqs. (78), (79) and (83). On the UNSAT side, our main results are Eqs. (85), (87), (88) and (101). From these results, the scaling of all the interesting observables can be derived. In this section, we focus on some of the most studied observables in the context of jamming, and we show that all the known results are reproduced by the scaling solution.

### 5.6.1 Energy, pressure, number of contacts

We start by considering the energy, the pressure and the number of contacts defined in Eq. (11). From Eqs. (88) and (101), we deduce that in the jamming limit from the UNSAT (jammed) phase,

$$\rho(h < 0) \sim \chi^2 p_-(\chi^2 h) \qquad \Rightarrow \qquad [h^n] \propto \chi^{-2n} . \quad (102)$$

In particular, the energy and pressure scale as

$$p = -[h] \sim \chi^{-2} \int_{-\infty}^0 dt\, p_-(t)|t| , \qquad e = \frac{\alpha}{2}[h^2] \sim \frac{\alpha}{2}\chi^{-4} \int_{-\infty}^0 dt\, p_-(t)t^2 \propto p^2 . \quad (103)$$

Note also that from Eq. (87) we have for the isostatic index:

$$c = \alpha z = \alpha[1] = \left(1 + \frac{1}{\chi}\right)^2 \sim 1 + \frac{2}{\chi} . \quad (104)$$

We deduce that *at jamming the system is isostatic*, and that the excess of contacts in the jammed phase scales like $c - 1 \propto \chi^{-1} \propto p^{1/2}$. To obtain the complete phenomenology of jamming, one should also prove that the pressure $p \propto \chi^{-2}$ vanishes linearly in the distance from jamming in the $(\alpha, \sigma)$ plane: we did not find a proof of this relation, which at present must be derived from the numerical solution of the equations. Apart from that, the scaling relations $e \propto p^2$ and $c - 1 \propto p^{1/2}$ perfectly agree with numerical observations in jammed sphere packings [5,6].

In the SAT phase, at zero temperature, there are by definition no negative gaps, $\rho(h < 0) = 0$; consequently pressure, energy, and contacts all vanish. However, one can study the limiting values of these quantities when $T \to 0$. As an example, let us consider the pressure, which is a standard observable in particle systems. In the perceptron, it can be written as the derivative of the free energy with respect to $\sigma$, as it can be checked directly from the definition of the partition function in Eq. (17). Using the replica expression of the free energy in Eq. (31), one then obtains:

$$
\begin{aligned}
p = -[h] &= \frac{1}{\alpha} \frac{\mathrm{d}f}{\mathrm{d}\sigma} \\
&= -T \frac{\mathrm{d}}{\mathrm{d}\sigma} \int \mathrm{d}h \gamma_{q_m}(h) f(0, h - \sigma) = T \int \mathrm{d}h \gamma_{q_m}(h) f'(0, h - \sigma) \\
&= T \int \mathrm{d}h P(0, h) f'(0, h) \,.
\end{aligned}
\tag{105}
$$

Using the equations for $P$ and $f$, it is possible to show that

$$
0 = \frac{\mathrm{d}}{\mathrm{d}x} \int \mathrm{d}h P(x, h) f'(x, h) \qquad \Rightarrow \qquad p = T \int \mathrm{d}h P(1, h) f'(1, h) \,.
\tag{106}
$$

The proof is obtained by writing the derivative as $\int \mathrm{d}h (\dot{P} f' + P \dot{f}')$, using the equations for $P$ and $f$ and integrating by parts [24]. Therefore, in the SAT phase the pressure vanishes proportionally to $T$, a standard result for hard spheres (which correspond to the $T \to 0$ limit of soft spheres in the SAT phase). However approaching the jamming line the ratio $p/T$ (also called "reduced pressure" or "compressibility factor" in the hard spheres literature) diverges. Inserting the scaling form given by Eq. (70), noting that $\lambda(1) = 1 - q_M = \epsilon^\kappa$, one has

$$
p = T \int \mathrm{d}h P(1, h) \frac{m(1, h)}{\lambda(1)} = \frac{T}{\epsilon^\kappa} \int \mathrm{d}h P(1, h) m(1, h) \,.
\tag{107}
$$

The scaling form in Eq. (70) is cutoff when $1 - q \sim \epsilon^\kappa$, and therefore the same scaling holds for $P(1, h)$ and $m(1, h)$ with $1 - q \to \epsilon^\kappa$. Then, in the $\epsilon \to 0$ limit the region $h > 0$ does not contribute to the integral because $m(1, h) \to 0$ while $P(1, h)$ stays finite. One can check that the matching region also gives a subdominant contribution. The leading term is associated to the $h < 0$ region, where $m(1, h) \to -h$ and

$$
p = \frac{T}{\epsilon^\kappa} \int_{-\infty}^{0} \mathrm{d}h \epsilon^{1-\kappa} p_-(h \epsilon^{1-\kappa}) |h| = \frac{T}{\epsilon} \int_{-\infty}^{0} \mathrm{d}t \, p_-(t) |t| \,.
\tag{108}
$$

We conclude that the reduced pressure $p/T$ diverges proportionally to $\epsilon^{-1}$, and therefore $1 - q_M \propto (p/T)^{-\kappa}$, a result that has been numerically tested in hard sphere systems [13]. This provides a physical meaning for the exponent $\kappa$. In a similar way one can study the scaling of the energy and the number of contacts in the SAT phase.

### 5.6.2 Force and gap distributions

In the UNSAT (jammed) phase, positive gaps correspond to satisfied constraints, while negative gaps can be associated to contact forces according to Eq. (12). The distribution of small positive gaps and small contact forces has been associated with important properties of the packings, including marginal stability [7, 8]. In this section we discuss these distributions. They can be straightforwardly derived from Eq. (88) and Eq. (101) which together imply, for $\chi \gg 1$:

$$
\rho(h) \sim \begin{cases} \chi^2 p_-(\chi^2 h) & \text{for } h < 0 \,, \\ p_+(h) & \text{for } h > 0 \,. \end{cases}
\tag{109}
$$

At jamming, when $\chi \to \infty$, the distribution of gaps concentrates on $h \geq 0$, where it is given by

$$\rho(h) = [1]\delta(h) + p_+(h) , \qquad h \geq 0 , \qquad (110)$$

where $[1] = \int_{-\infty}^{0} dh \rho(h) = \int_{-\infty}^{0} dt p_-(t) = 1/\alpha$ according to Eq. (104). Isostaticity is expressed by the presence of a delta peak in $\rho(h)$ whose weight is the fraction of contacts. Moreover, this implies that the distribution of small positive (satisfied) gaps, according to Eq. (78), behaves as a power-law $\rho(h \to 0^+) \sim h^{-\gamma}$ with $\gamma = (2-\kappa)/\kappa = 0.41269\dots$. Similarly, the normalized probability distribution of scaled contact forces $f_\mu^s = -[1]h_\mu/p$, with $p/[1]$ given by Eq. (103), can be deduced from $\rho(h < 0)$ and reads

$$p(f) = \alpha \bar{f} p_-(-f\bar{f}) , \qquad \bar{f} = \alpha \int_0^\infty dt \, p_-(-t) t , \qquad \frac{1}{\alpha} = [1] = \int_0^\infty dt \, p_-(-t) . \quad (111)$$

Note that $\int_0^\infty df \, p(f) = \int_{-\infty}^0 df \, p(f) f = 1$. This implies, according to Eq. (79), that the distribution of small forces behaves as a power-law, $p(f \to 0^+) \sim f^\theta$, with $\theta = 1/\gamma - 2 = 0.42311\dots$. These results give a physical interpretation to the critical exponents $\gamma$ and $\theta$, and connect with the results of [7–10, 12]. Similar results can be obtained upon approaching the jamming transition from the SAT phase, with the only difference that in that case one has to define the forces by separating the positive gaps into those who become contacts at jamming and those who stay positive [9, 25, 38]. As a check of the theoretical predictions we have extended the numerical simulations of [14] to large systems: in Fig. 4 we compare the cumulative distribution $C_f = \int_0^f d\tilde{f} p(\tilde{f})$ for sizes from $N = 50$ to $N = 1600$ with the theoretical prediction.

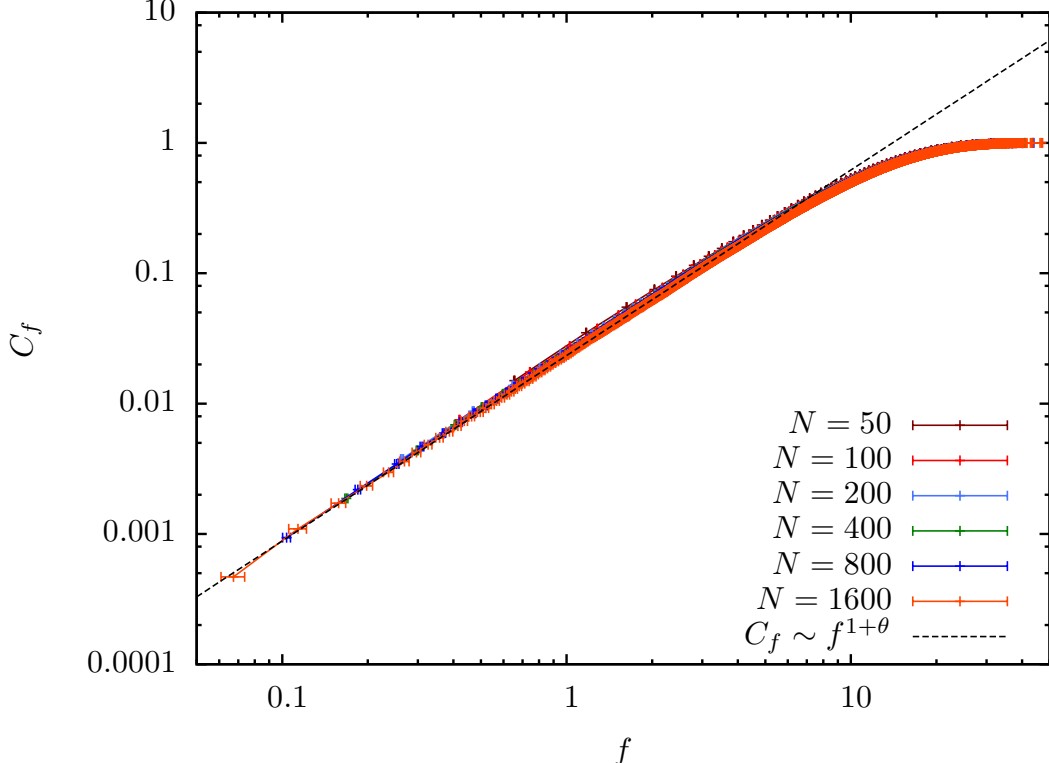

Figure 4: Plot of the cumulative of the force distribution $C_f = \int_0^f d\tilde{f} p(\tilde{f})$ for $N = 50, \dots 1600$. The line is proportional to the theoretical prediction $C_f \sim f^{1+\theta}$.

We conclude by mentioning that one can also study the vibrational spectrum, both in the SAT and UNSAT phases [16, 17], to discuss the presence of soft modes associated to both fullRSB and jamming, and connect the density of states with these critical exponents as it is also observed in sphere packings [40]. Also, it has been shown in [19] that the asymptotic behavior of the function $y(q)$, related to the exponent $\kappa$, controls the scaling of the avalanches at the jamming point, which is therefore different than the scaling in the UNSAT phase.

# 6 Conclusions

In this work we have formulated the jamming problem as a general constraint satisfaction problem with continuous variables (Sec. 2). We have then specialized on the random perceptron, a well known machine learning model, which is a prototype of this class of problems (Sec. 3). In the non-convex regime, the model shows a complex zero temperature phase diagram in the plane of the two control parameters $(\alpha, \sigma)$, which has been fully characterized in Sec. 4. In particular, we have shown that for $\sigma < 0$ and large enough $|\sigma|$, the phase diagram as a function of $\alpha$ shows the characteristic phenomenology associated with the Random First Order Transition (RFOT) mean field theory of glasses [27, 33–36]. Our main result is that the jamming transition, which can be seen as the SAT/UNSAT threshold, is always associated with full replica symmetry breaking in the non-convex regime. In Sec. 5, we have thus discussed the scaling behavior of the fullRSB equations around the jamming transition. We have shown that approaching jamming from the SAT phase, one obtains the same critical exponents of the jamming transition of hard spheres in high dimension, thus reproducing the results of [13]. Furthermore we have extended the study of [13] by analyzing the model in the UNSAT phase, where we have obtained the scaling solution of the fullRSB equations, showing that it reproduces the critical behavior of soft spheres, when approaching the transition from the jammed phase [5]. We have provided a complete matching between the scaling solutions in the SAT and UNSAT phases, and derived scaling relations between the critical exponents, also showing their physical interpretation as the exponents that control the gap and force distributions at jamming [7]. Our results are also consistent with the scaling analysis of [11]. These results, together with the results on the vibrational spectrum obtained in [16, 17], and the study of avalanches performed in [19], provide a complete study of all the properties of the random perceptron that are relevant for the study of the glass and jamming transition at the mean field level.

This work opens the way to the study of the jamming transition in other ensembles of random constraint satisfaction problems with continuous variables. The outcome of this study is that the non-convex jamming transition lies always in a fullRSB phase and we conjecture that this happens in a large class of CCSP. Although we are not able to prove it we note that this is what happens not only in the model we have analyzed but also in Hard-Spheres in high dimension [13]. Another natural question that arises is to what extent the scaling behavior that we have found is universal. The fact that the critical exponents at the jamming transition for the random perceptron and hard spheres coincide support the conjecture that in non-convex random CSPs, the jamming point is highly universal. This conjecture has been tested and confirmed in generalized perceptron models with multibody interaction [41]. There could be, however, different universality classes: understanding which features of the models determine them is a very important direction for future work. Going beyond the mean field, infinite dimensional level, for example by considering random dilute versions of the perceptron, is another interesting direction for future research.

## Acknowledgements

This work was supported by grants from the Simons Foundation (No. 454941, S. F.; No. 454949, G. P.; No. 454955, F. Z.). This work is supported by "Investissements d'Avenir" LabEx PALM (ANR-10-LABX-0039-PALM) (P. Urbani).

## A  Derivation of the replicated free energy

In this Appendix, we give a detailed derivation of the replica equations for the perceptron model. Starting from the partition function in Eq. (17), and neglecting proportionality constants, the replicated partition function can be written as

$$\overline{Z^n} \propto \int \left(\prod_a \mathscr{D}\vec{X}_a\right)\left(\prod_{a,\mu} dr_a^\mu d\hat{r}_a^\mu\right)\overline{e^{\sum_{a\mu} i\hat{r}_a^\mu(r_a^\mu - N^{-1/2}\vec{X}_a\cdot\vec{\xi}^\mu)}}\prod_{a\mu} e^{-\beta v(r_a^\mu - \sigma)} \,, \tag{112}$$

where $a = 1\cdots n$ and $\mu = 1\cdots M$; one can check that integrating over $\hat{r}_a^\mu$ produces delta functions that fix $r_a^\mu$ as in the original partition function. Next, the Gaussian integral over the quenched disorder $\vec{\xi}^\mu$ gives

$$\overline{e^{-N^{-1/2}\sum_{a\mu} i\hat{r}_a^\mu\vec{X}_a\cdot\vec{\xi}^\mu}} = e^{-\frac{1}{2}\sum_{ab\mu}\hat{r}_a^\mu\hat{r}_b^\mu Q_{ab}} \,, \qquad Q_{ab} = \frac{1}{N}\vec{X}_a\cdot\vec{X}_b \,. \tag{113}$$

We obtain then

$$\overline{Z^n} \propto \int \left(\prod_a \mathscr{D}\vec{X}_a\right)\left[\int \left(\prod_a dr_a d\hat{r}_a\right)e^{\sum_a i\hat{r}_a r_a - \frac{1}{2}\sum_{ab}\hat{r}_a\hat{r}_b Q_{ab} - \beta\sum_a v(r_a-\sigma)}\right]^M$$

$$\propto \int dQ_{ab} e^{\frac{N}{2}\log\det Q}\left[\int \left(\prod_a dr_a d\hat{r}_a\right)e^{\sum_a i\hat{r}_a r_a - \frac{1}{2}\sum_{ab}\hat{r}_a\hat{r}_b Q_{ab} - \beta\sum_a v(r_a-\sigma)}\right]^M \,, \tag{114}$$

where we made a change of variables from $\vec{X}_a$ to $Q_{ab}$, with Jacobian $\exp(\frac{N}{2}\log\det Q)$. Finally, one can easily show by developing both sides in powers of $Q_{ab}$ that

$$e^{-\frac{1}{2}\sum_{ab}\hat{r}_a\hat{r}_b Q_{ab}} = \left.e^{\frac{1}{2}\sum_{ab} Q_{ab}\frac{\partial^2}{\partial k_a\partial k_b}}e^{-\sum_a k_a i\hat{r}_a}\right|_{k_a=0} \,, \tag{115}$$

and therefore

$$\int \left(\prod_a dr_a d\hat{r}_a\right)e^{\sum_a i\hat{r}_a r_a - \frac{1}{2}\sum_{ab}\hat{r}_a\hat{r}_b Q_{ab} - \beta\sum_a v(r_a-\sigma)} \tag{116}$$

$$= \left.\int \left(\prod_a dr_a d\hat{r}_a\right)e^{\frac{1}{2}\sum_{ab} Q_{ab}\frac{\partial^2}{\partial k_a\partial k_b}}e^{\sum_a(r_a-k_a)i\hat{r}_a - \beta\sum_a v(r_a-\sigma)}\right|_{k_a=0} \tag{117}$$

$$\propto \left.e^{\frac{1}{2}\sum_{ab} Q_{ab}\frac{\partial^2}{\partial k_a\partial k_b}}e^{-\beta\sum_a v(k_a-\sigma)}\right|_{k_a=0} \,, \tag{118}$$

where to obtain the last line we integrated over $\hat{r}_a$ to obtain a $\delta(r_a - k_a)$ and then integrated over $r_a$. Introducing $h_a = k_a - \sigma$ and setting $\alpha = M/N$, we obtain the final result for the replicated partition function

$$\overline{Z^n} = \int dQ e^{NS(Q)} \,, \tag{119}$$

$$S(Q) = \frac{1}{2}\log\det Q + \alpha\log\left(\left.e^{\frac{1}{2}\sum_{ab} Q_{ab}\frac{\partial^2}{\partial h_a\partial h_b}}\prod_a e^{-\beta v(h_a)}\right|_{h_a=-\sigma}\right) \,, \tag{120}$$

that is reported in Eq. (21).

# B  Numerical solution of the equations

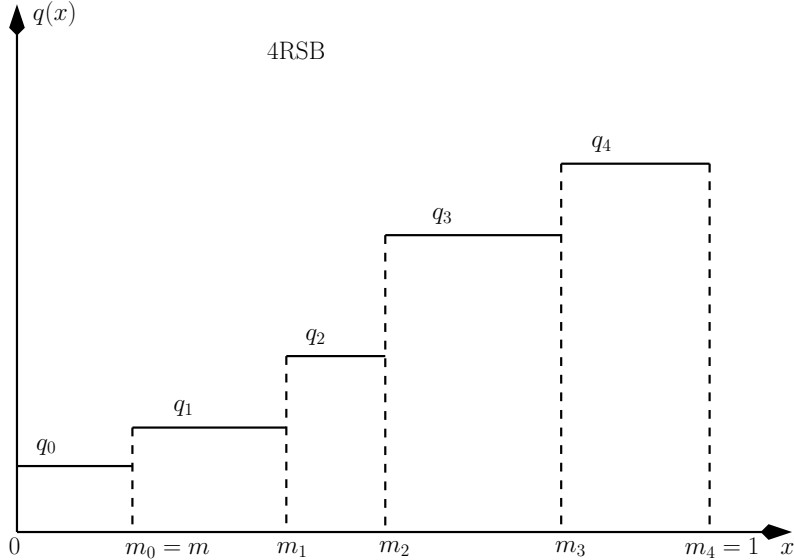

Figure 5: An example of the parametrization of the matrix $Q_{ab}$ for a 4RSB case. The reader should keep in mind that we use the convention $m_0 \equiv m$, $m_{k+1} \equiv 1$.

In this Appendix, we derive the equations corresponding to a discrete $K$RSB ansatz, illustrated in Fig. 5. This is particularly useful for the numerical solution.

For a function $q(x)$ with $K$RSB form, the free energy of Eq. (31) is

$$s[q(x)] = \lim_{n \to 0} \frac{1}{n} S(Q) = \frac{1}{2} \left[ \log(1 - \langle q \rangle) + \frac{q_m}{1 - \langle q \rangle} + \int_0^1 \frac{dy}{y^2} \log\left(\frac{\lambda(y)}{1 - \langle q \rangle}\right) \right] + \alpha f_S(-\sigma) \quad (121)$$

with $\langle q \rangle = \int_0^1 dx\, q(x)$, and $\lambda(x)$ is also a piecewise constant function whose values can be derived from $q(x)$. The function $f(m_i, h)$ is the discrete version of $f(q, h)$ and it satisfies the discrete version of Eq. (30):

$$\begin{cases} f(1, h) = \log \gamma_{1-q_k} \star e^{-\beta v(h)}\,, \\ f(m_i, h) = \frac{1}{m_i} \log\left[\gamma_{q_{i+1} - q_i} \star e^{m_i f(m_{i+1}, h)}\right]\,, & i = k-1, \cdots, 0\,, \\ f_S(h) = \gamma_{q_0} \star f(m_0, h)\,. \end{cases} \quad (122)$$

The saddle point equation is

$$Q_{cd}^{-1} = -\alpha \frac{\left( e^{\frac{1}{2} \sum_{ab} Q_{ab} \frac{\partial^2}{\partial h_a \partial h_b}} \frac{\partial^2}{\partial h_c \partial h_d} \prod_a e^{-\beta v(h_a)} \Big|_{h_a = -\sigma} \right)}{\left( e^{\frac{1}{2} \sum_{ab} Q_{ab} \frac{\partial^2}{\partial h_a \partial h_b}} \prod_a e^{-\beta v(h_a)} \Big|_{h_a = -\sigma} \right)} \equiv -\alpha M_{cd}\,, \quad (123)$$

and the matrix $M_{ab}$ is also a hierarchical matrix whose components can be written as

$$M_i = \int dh P(m_i, h) f'(m_i, h)^2\,, \quad (124)$$

with

$$\begin{cases} P(m_0, h) = \gamma_{q_0}(h + \sigma) \\ P(m_i, h) = e^{m_{i-1} f(m_i, h)} \gamma_{q_i - q_{i-1}} \star \left[P(m_{i-1}, h) e^{-m_{i-1} f(m_{i-1}, h)}\right]\,, & i = 1, \cdots, k\,, \end{cases} \quad (125)$$

which provides a discrete version of Eq. (34). Note that all the $P(m_i, h)$ are normalized to 1. The saddle point equation for a hierarchical matrix are therefore

$$q_i^{-1} = -\alpha M_i = -\alpha \int \mathrm{d}h P(m_i, h) f'(m_i, h)^2 . \tag{126}$$

To close the equations, we now need to express $q_i$ as a function of $q_i^{-1}$.

We start from the exact relations (note that here $q_d = 1$ is the diagonal element and $[q](x) = x q(x) - \int_0^x \mathrm{d}y q(y)$):

$$\lambda(x) \equiv q_d - x q(x) - \int_x^1 \mathrm{d}y q(y) \quad \Leftrightarrow \quad q(x) = q_d - \frac{\lambda(x)}{x} + \int_x^1 \frac{\mathrm{d}y}{y^2} \lambda(y) ,$$

$$(q^{-1})_d = \frac{1}{\lambda(0)} \left( 1 - \int_0^1 \frac{\mathrm{d}y}{y^2} \frac{[q](y)}{\lambda(y)} - \frac{q(0)}{\lambda(0)} \right) , \tag{127}$$

$$(q^{-1})(x) = -\frac{1}{\lambda(0)} \left[ \frac{q(0)}{\lambda(0)} + \frac{[q](x)}{x \lambda(x)} + \int_0^x \frac{\mathrm{d}y}{y^2} \frac{[q](y)}{\lambda(y)} \right] = -\frac{q(0)}{\lambda(0)^2} - \int_0^x \frac{\mathrm{d}y}{\lambda(y)^2} \dot{q}(y) .$$

The above equalities are obtained by showing that the derivatives with respect to $x$, as well as the values in $x = 0$ or $x = 1$ coincide. From these one can derive several useful identities:

$$(q^{-1})_d - \langle q^{-1} \rangle - [q^{-1}](x) = \frac{1}{q_d - \langle q \rangle - [q](x)} \quad \Rightarrow \quad (q^{-1})_d - \langle q^{-1} \rangle = \frac{1}{q_d - \langle q \rangle} . \tag{128}$$

From the last Eq. (127) we get $(q^{-1})(0) = -\frac{q(0)}{\lambda(0)^2}$. Collecting these two relations we get

$$\lambda(0) = \sqrt{-\frac{q(0)}{(q^{-1})(0)}} , \qquad \frac{1}{\lambda(x)} = \frac{1}{\lambda(0)} - [q^{-1}](x) . \tag{129}$$

This relations allows one to reconstruct $\lambda(x)$ from $(q^{-1})(x)$ and using the first Eq. (127) we can obtain $q(x)$ from $\lambda(x)$. In the discrete case one has $[q]_i = \sum_{j=0}^{i-1} m_j (q_{j+1} - q_j)$ and then these equations become

$$\lambda_0 = \sqrt{-\frac{q_0}{q_0^{-1}}} , \qquad \frac{1}{\lambda_i} = \frac{1}{\lambda_{i-1}} - m_{i-1}(q_i^{-1} - q_{i-1}^{-1}) .$$

$$q_i = 1 - \frac{\lambda_i}{m_i} - \sum_{j=i+1}^{k} \left( \frac{1}{m_j} - \frac{1}{m_{j-1}} \right) \lambda_j . \tag{130}$$

The procedure to solve these equations is therefore (for a fixed grid of $m_i$): *(i)* start from a guess for $q_i$, *(ii)* solve Eqs. (122) and (125) to obtain $f(m_i, h)$ and $P(m_i, h)$, *(iii)* from Eq. (126) compute $q_i^{-1}$, and *(iv)* use Eqs. (130) to compute the new $q_i$.

## C  Asymptotic behavior in the UNSAT phase

In this Appendix we discuss the asymptotic behavior of $f(q_M, h)$ in the UNSAT phase where $q_M = 1 - \chi T$. This is given by the zero temperature limit of

$$f(q_M, h) = \log \int_{-\infty}^{\infty} \frac{\mathrm{d}z}{\sqrt{2\pi \chi T}} \exp \left[ -\frac{(h-z)^2}{2\chi T} - \frac{z^2}{2T} \theta(-z) \right] . \tag{131}$$

For $T \to 0$ and $h \sim \mathcal{O}(1)$ we can compute the integral using a saddle point approximation. The saddle point equation is

$$\frac{z-h}{\chi} + z\theta(-z) = 0 \,. \tag{132}$$

Its solutions are given by

$$
\begin{aligned}
z^*(h) &= h \,, &&\text{for } h > 0 \,, \\
z^*(h) &= \frac{h}{1+\chi} \,, &&\text{for } h < 0 \,.
\end{aligned}
\tag{133}
$$

Thus, defining $t = z - z^*(h)$ obtain

$$
\begin{aligned}
f(q_M, h) &= \log\left[\theta(h)\int_{-\infty}^{\infty}\frac{\mathrm{d}t}{\sqrt{2\pi\chi T}}e^{-t^2/(2\chi T)} + \theta(-h)e^{-h^2/(2(1+\chi)T)}\int_{-\infty}^{\infty}\frac{\mathrm{d}t}{\sqrt{2\pi\chi T}}e^{-\frac{1+\chi}{2\chi T}t^2}\right] \\
&\simeq \left(-\frac{h^2}{2(1+\chi)T} - \frac{1}{2}\ln(1+\chi)\right)\theta(-h) \,.
\end{aligned}
\tag{134}
$$

# D  Scaling of the pressure in the UNSAT phase

In this Appendix, we prove that the pressure satisfies the relation $p = -[h] \propto 1/\chi^2$, a relation used to prove the scaling relation $a = 1 - \kappa/2$ in Sec. 5.5. To this aim, let us modify the problem by replacing the hard spherical constraint by a Lagrange multiplier $\mu$, and consider the following exact relation

$$0 = \frac{1}{ZN}\int \mathrm{d}\vec{X}\sum_{i=1}^{N}\frac{\partial}{\partial X_i}\left\{X_i\exp\left[-\beta\left(H[\vec{X}] - \frac{\mu}{2}\sum_{j=1}^{N}(X_i^2 - 1)\right)\right]\right\} \,, \tag{135}$$

from which it follows that

$$\mu = -T + [h^2] + \sigma[h] \,. \tag{136}$$

We will prove below that in the zero temperature limit and in the fullRSB phase, $\mu = 1/\chi^2$, so that the following exact zero temperature relation holds:

$$1 = \chi^2\left([h^2] + \sigma[h]\right) \,. \tag{137}$$

Because close to jamming $[h^2] \ll [h]$ this reduces to $\lim_{\chi \to \infty}\chi^2\sigma[h] = 1$ at jamming, which proves $[h] \propto 1/\chi^2$.

We now compute $\mu$ using the replica method. Because $X^2$ is not constrained, we have that $Q_{aa} = q_d$ and we need to find an additional variational equation for $q_d$. Let us write down the free energy as a function of $q_d$ and $\mu$. We have

$$
\begin{aligned}
s[q_d, q(x)] ={}& \frac{1}{2}\left[\log(q_d - q_M) + \frac{q_m}{q_d - \langle q\rangle} + \int_0^1 \mathrm{d}x\,\frac{\dot{q}(x)}{\lambda(x)}\right] + \alpha\gamma_{q_m} \star f(0, h)|_{h=-\sigma} + \frac{1}{2}\beta\mu(q_d - 1) \\
&- \alpha\int \mathrm{d}h\, P(q_M, h)\left[f(q_M, h) - \log\gamma_{q_d - q_M} \star e^{-\beta v(h)}\right] \\
&+ \alpha\int \mathrm{d}h\int_{q_m}^{q_M}\mathrm{d}q\, P(q, h)\left[\dot{f}(q, h) + \frac{1}{2}\left[f''(q, h) + x(q)f'(q, h)^2\right]\right].
\end{aligned}
\tag{138}
$$

where now

$$\lambda(x) = q_d - x q(x) - \int_x^1 \mathrm{d}y\, q(y)$$
$$\lambda(q) = q_d - q_M + \int_q^{q_M} \mathrm{d}p\, x(p) \,. \tag{139}$$

The variational equation for $q(x)$, $f(q,h)$ and $P(q,h)$ do not change except for the initial condition for $f$ which is

$$f(q_M, h) = \log \gamma_{q_d - q_M} \star e^{-\beta v(h)} \tag{140}$$

and $\lambda(q)$ is given by Eq. (139). Note that the variational equation with respect to $\mu$ fixes the spherical constraint $q_d = 1$. At this point we can take the variation with respect to $q_d$ to get the following equation

$$0 = \frac{1}{2}\left[\frac{1}{q_d - q_M} - \frac{q_m}{(q_d - \langle q\rangle)^2} - \int_0^1 \mathrm{d}u\, \frac{\dot{q}(u)}{\lambda^2(u)}\right] + \alpha \int \mathrm{d}h P(q_M, h)\frac{\partial}{\partial q_d}\log\gamma_{q_d - q_M}\star e^{-\beta v(h)} + \frac{\beta\mu}{2}\,. \tag{141}$$

It is very easy to show that the last term of the equation above can be rewritten as

$$\alpha \int \mathrm{d}h P(q_M, h)\frac{\partial}{\partial q_d}\log\gamma_{q_d - q_M}\star e^{-\beta v(h)} = \frac{\alpha}{2}\int \mathrm{d}h P(q_M, h)\left[m'(q_M, h) + m^2(q_M, h)\right] \tag{142}$$

where again $m(q_M, h) = \partial f(q_M, h)/\partial h$. At this point we can use the saddle point equations (35) and (36) to write

$$\frac{q_m}{\lambda^2(q_m)} + \int_{q_m}^{q_M}\frac{\mathrm{d}p}{\lambda^2(p)} = \alpha \int \mathrm{d}h P(q_M, h) m^2(q_M, h)\,,$$
$$\frac{1}{(q_d - q_M)^2} = \alpha \int \mathrm{d}h P(q_M, h)\left(m'(q_M, h)\right)^2 \,. \tag{143}$$

Using that the variational equation over $\mu$ gives $q_d = 1$, that $q_M = 1 - \chi T$ and that

$$m'(q_M, h) = -\frac{\beta}{1+\chi}\theta(-h) \tag{144}$$

we get $\mu = 1/\chi^2$.

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
