# Peer review of "Universality of the SAT-UNSAT (jamming) threshold in non-convex continuous constraint satisfaction problems"

_SciPost Physics, doi:SciPost Phys. 2, 019 (2017)_

## Round 1 · Referee Report · Anonymous (Referee 1) · 2017-4-13

Strengths

1- brings together two fields, jamming and constraint satisfaction problems (CSP) with continuous variables

2- paper is complete, detailed and very well written

3- show signatures of universalities in mean field jamming/CSP transitions

4- phase diagram and exponents computed exactly

Weaknesses

1 - The paper is very technical, perhaps a little bit difficult to read by non experts. However all the pointers are given to the readers.

Report

This is an impressive paper which builds a complete and deep connection between jamming transitions and sat/unsat transitions in random constraint satisfaction problems (CSP) with continuous interactions.
The paper is extremely well written. It provides all the physical insights and definitions of the jamming observables which are needed to interpret the full phase diagram of CSP problems in physical terms. The sphere packing problem in high dimension is represented as a perceptron model with appropriate (negative) stability. The corresponding packing problem is highly non convex as in jamming. Quite remarkably the full T=0 phase diagram is derived, including all levels of symmetry breaking.
The analysis is taken to its limits by computing the scaling solution at the SAT phase boundary of the perceptron. The scaling relations between critical exponents is interpreted in terms of gaps and force distribution at the jamming point.

In my opinion, this paper deserves publication in its present form. There is no doubt that it provides an important unifying perspective on jamming transitions at mean-field level. Moreover is also provides new insight for random CSPs with continuous variables.

The fact that the perceptron model and hard spheres have the same mean field exponents (in infinite dimension) is important. The authors outline some interesting perspectives for future studies aimed at establishing the degree of universality of these results.

Requested changes

1- no changes

---

## Round 1 · Referee Report · Anonymous (Referee 2) · 2017-4-25

Strengths

It looks like an exemplar piece of theoretical work

Weaknesses

Probably is connected to its strength. It's too technical and hard to follow for somebody without a background in disordered systems.

Report

I must admit that I didn't have the time to properly check all the results presented by the authors, although they look to me reasonable, physically sound and feasible to be reproduced with the technical information provided in the manuscript.

The paper is very well written, the results are original and it probably concludes the analysis of the thermodynamics of the problem already started by the authors and other collaborators in ref's [14], [16]-[19]. Therefore I strongly recommend this work as an important piece in the study of disordered systems with continuous variables.

Requested changes

1-However, I would like to suggest to the authors to make clearer in the conclusions that their statement that the SAT/UNSAT transition is fullRSB, is a statement valid strictly for the model studied in the manuscript and not necessarily valid for other non-convex continuous CSP. At least, in the way that I understood the manuscript. If this is not the case, it is worth to say a few words about why the authors believe this statement is more general than that.

  • validity: top
  • significance: high
  • originality: top
  • clarity: high
  • formatting: perfect
  • grammar: excellent

Author:  Pierfrancesco Urbani  on 2017-04-29  [id 123]

(in reply to Report 2 on 2017-04-25)
Category:
remark

We thank the referee for her/his report. We have followed her/his suggestion and we have added a sentence in the conclusions underlining what she/he was mentioning. We are not able to prove that the non-convex jamming transition is always in a fullRSB phase. Nevertheless there are at least two models where this is the case that are the random perceptron and hard spheres in high dimension. We resubmit an updated version of the paper with very few typos corrected.

---

## Round 2 · List of Changes

We have corrected a typo in Eq. (30) (wrong sign in front of the integral). We have corrected the same typo wherever it appeared in the text.
We have also corrected a typo in Eq. (D4) where there was a missing 1/2 factor in the first line.
Finally we have included a referee's suggestion in the conclusions.

---

## Editorial Decision

published